# TACTICS OF ROBUST DEEP REINFORCEMENT LEARNING WITH RANDOMIZED SMOOTHING

## ABSTRACT

Despite randomized smoothing being proven to give a robustness guarantee, the standard performance of a smoothed deep reinforcement learning (DRL) agent exhibits a significant trade-off between its utility and robustness. Naively introducing randomized smoothing during the training or testing can fail completely in the DRL setting. To address this issue, we proposed new algorithms to train smoothed robust DRL agents while attaining superior clean reward, empirical robustness, and robustness guarantee in discrete and continuous action space. Our proposed DS-DQN and AS-PPO outperform prior state-of-the-art robustly-trained agents in robust reward by $1.6\times$ on average and exhibit strong guarantees that previous agents failed to achieve. Moreover, a stronger adversarial attack for smoothed DQN agents is proposed, which is $4.6\times$ more effective in decreasing the rewards compared to existing adversarial attacks.

## 1 INTRODUCTION

Deep Reinforcement Learning (DRL) has reached performance beyond humans in many game environments (Mnih et al., 2013; Silver et al., 2016) as well as many safety-critical domains, such as robotics (Kober et al., 2013; Fisac et al., 2019), autonomous driving (Kiran et al., 2022) and healthcare (Yu et al., 2023). Unfortunately, recent studies pointed out that DRL is vulnerable to adversarial perturbations (Huang et al., 2017; Lin et al., 2017; Weng et al., 2020). Hence, it is necessary to improve the robustness of the DRL agents before they can be deployed to real-world applications, especially for safety-critical tasks.

Recently, the techniques developed for training a robust classifier have been adapted to improve the robustness of DRL agents. For example, Pattanaik et al. (2018) adopted adversarial training (Madry et al., 2018; Yuan et al., 2019) to train DRL agents with adversarial examples, and Zhang et al. (2020); Oikarinen et al. (2021) proposed to robustify DRL agents with regularizers based on robustness verification bounds (Gowal et al., 2018). More recently, Wu et al. (2022) proposed the first framework named CROP to transform a DRL agent to a smoothed agent via Randomized Smoothing (RS) (Cohen et al., 2019). CROP provided certified robustness guarantees for discrete-action agents (e.g. DQN), and they showed that the certified radius of a smoothed agent is generally larger compared to the vanilla agents when the original base agent is trained via robust training (e.g. SADQN (Zhang et al., 2020), RadialDQN (Oikarinen et al., 2021)). Their approach does not involve any training on the smoothed agents, as the transformation only involves applying RS.

Nevertheless, we found that the pipeline in CROP (Wu et al., 2022) may have a few issues. First, there exists a significant trade-off between the clean reward and the robustness of the CROP agents. The CROP agents cannot tolerate the large noise introduced by RS and suffer from a significant decrease in clean reward. In other words, CROP agents are not usable despite being robust. We present our detailed observation and discussion in Section 2 *Failure in existing smoothed DRL agents*. Second, the smoothing strategy in CROP may lead to an overestimation of the certifiable robustness. Similarly, the attack evaluation is also ineffective in decreasing the reward of smoothed agents, which may also provide an illusion of empirical robustness. We will discuss this in Section 3 *Issues of the smoothing strategy in CROP* and *Our stronger attack*.

Motivated by these limitations, in this work, our primary goal is to devise new methods to mitigate the trade-off between clean reward and robustness of the CROP agents and fix the issue of overestimation of the robustness with a new smoothing strategy and stronger attack. We first show that it

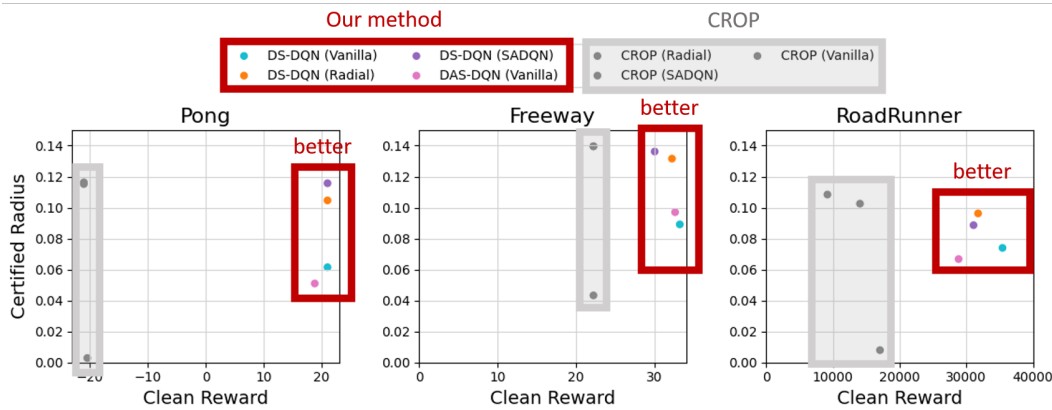

Figure 1: The average clean reward and certified radius of our DS-DQN and CROP (Wu et al., 2022). Our robust agents have much better clean reward compared to the CROP framework.

could be challenging to directly apply RS during the DRL training: the smoothed DRL agents could completely fail unlike the classification settings (Cohen et al., 2019), suggesting the necessity of more investigation and careful designs. Consequently, we propose two new smoothed DRL agents, DS-DQN and AS-PPO, with better smoothing strategies that can attain both great clean reward and robustness certification for discrete and continuous actions. Our agents establish new state-of-the-art results under our stronger attack setting across several standard RL benchmarks on Atari games (Brockman et al., 2016) and continuous control tasks (Brockman et al., 2016). Our contributions are summarized as follows:

- We identify the issues in CROP (Wu et al., 2022), showing the poor trade-off (see Section 2) and pointing out the overestimated robustness in their framework. We solve these issues by designing new training algorithms, introducing better smoothing strategies, and evaluating smoothed agents with our stronger attack. Furthermore, We extend the robust guarantee from DQN agents to the PPO setting and defined action bound for continuous-action agents (see Section 4).

- We develop the first robust DRL training algorithms leveraging randomized smoothing with other techniques for both discrete actions (DS-DQN) and continuous actions (AS-PPO). We also show that naively training with RS does not work and it is necessary to combine denoise smoothing and adversarial training.

- Our agents are the first state-of-the-art robust agents with a high robustness guarantee at the same time (certified radius and reward lower bound), while the previous state-of-the-art only evaluate their agents under empirical attacks. Our DS-DQN and AS-PPO earn $2.07\times$ and $1.25\times$ more reward respectively than the current best agents under the strongest attack.

## 2 FAILURE IN EXISTING SMOOTHED DRL AGENTS

Despite RS being amenable to providing robustness certification, we found that there is a **significant trade-off between the reward and the robustness of the smoothed agents** in CROP (Wu et al., 2022). In their approach, they evaluated smoothed agents with a large smoothing factor $\sigma = 0.1$ since it can increase the certified radius, which ensures the action of the smoothed agent remains unchanged within this radius. However, as we show in Figure 1, the clean reward of the CROP agents is degraded significantly, regardless the base DRL agents are robustly trained (RadialDQN, SADQN) or not (VanillaDQN). This makes the CROP agents impractical to use: a large certified radius is not useful as a robust but badly performed agent is not acceptable for deployment.

On the other hand, one may wonder: why not use a smaller $\sigma$ to avoid harming the performance of the smoothed agents? As we show in Figure 6 (Appendix A.5) the cost of maintaining good clean reward is to have much *weaker robustness*: for CROP, the certified radius is extremely small with small $\sigma$. Similar issues can be found in the smoothed agents with continuous actions (e.g. PPO) albeit less severe than the DQNs. The smoothed PPO agents (e.g. VanillaPPO+RS and SAPPO+RS

(SGLD) (Zhang et al., 2020)) have worse clean reward than their non-smoothed versions, which will be discussed in Section 5 Table 3.

In contrast, as shown in Figure 1, our proposed methods are capable of attaining high robustness and clean reward under a large $\sigma$ for both DQN and PPO agents, suggesting it is possible to mitigate the trade-off between the robustness and utility of smoothed DRL agents. In the following section, we introduce our proposed methods: DS-DQN for discrete actions and AS-PPO for continuous actions.

## 3 Learning robust DRL agents with randomized smoothing

In this section, we propose methods that leverage RS with other techniques to obtain certifiably robust agents, while mitigating the trade-offs mentioned above and fixing the pipeline of robustness evaluation in CROP. We focus on two representative RL algorithms: DQN for discrete action space, and PPO for continuous action space, which are the focus of prior works in robust DRL literature (Zhang et al., 2020; Oikarinen et al., 2021).

### 3.1 DS-DQN (Denoised Smoothed - Deep Q Network)

**Motivating example.** Based on the experiments of Figure 1 in Section 2, we found that none of the smoothed DQN agents in CROP (Wu et al., 2022) can tolerate the large Gaussian noise introduced by RS during the testing. To eliminate the inconsistency between training and testing, our first attempt is to incorporate RS into training, and we call this approach Smoothed-DQN (S-DQN). We formulate the temporal difference loss of S-DQN as $\mathcal{L}_{\text{TD}} = \mathbb{E}[(r + \gamma \max_{a'} Q_{\text{target}}(s', a') - Q(\tilde{s}, a; \theta))^2]$, where $\tilde{s} = s + \mathcal{N}(0, \sigma^2 I_N)$, $\sigma$ is the smoothing variance, $Q_{\text{target}}$ is the pretrained Q-Network, and $\theta$ is the parameters of S-DQN. This implementation makes S-DQN agents learn under noises. However, we found that S-DQN agents get very low clean reward despite having large certified radius as shown in Table 2 row (b) in Section 5. This suggests that simply involving RS in DQN training is not a useful idea, unlike the supervised learning setting where training with RS can achieve high clean accuracy (Cohen et al., 2019) for a classifier. Hence, it is important to develop a strategy to remove the noise from RS, which motivates us to propose DS-DQN leveraging the technique of Denoised Smoothing (Salman et al., 2020). We describe the details of training, testing, and evaluating our DS-DQN in the following paragraphs.

**Training and loss function.** The flow chart of the training process of DS-DQN is shown in Appendix A.2 Figure 4 (a). There are two parts of the training: collecting transitions and updating the networks. First, we collect the transitions $\{s_t, a_t, r_t, s_{t+1}\}$ by taking the $\epsilon$-greedy strategy, which can be formulated as follows:

$$a_t = \begin{cases} \arg\max_a Q(D(\tilde{s}_t; \theta), a), & \text{with probability } 1 - \epsilon \\ \text{Random Action}, & \text{with probability } \epsilon \end{cases} \tag{1}$$

where $D$ is the denoiser that removes the noise from the input states, $Q$ is the pretrained Q-network, $\tilde{s}_t$ is the state with noise $\tilde{s}_t = s_t + \mathcal{N}(0, \sigma^2 I_N)$, and $\sigma$ is the standard deviation of the Gaussian distribution. After collecting the transitions, they are stored in the replay buffer. In the second stage, we sample some transitions from the replay buffer and update the parameters of the denoiser $D$. The entire loss function is designed with two parts — reconstruction loss $\mathcal{L}_{\text{R}}$ and temporal difference loss $\mathcal{L}_{\text{TD}}$:

$$\mathcal{L} = \lambda_1 \mathcal{L}_{\text{R}} + \lambda_2 \mathcal{L}_{\text{TD}}, \tag{2}$$

where $\lambda_1$ and $\lambda_2$ are the hyperparameters. Suppose the sampled transition is $\{s, a, r, s'\}$, the reconstruction loss $\mathcal{L}_{\text{R}}$ is defined as:

$$\mathcal{L}_{\text{R}} = \frac{1}{N} ||D(\tilde{s}; \theta) - s||_2^2, \tag{3}$$

where $\tilde{s} = s + \mathcal{N}(0, \sigma^2 I_N)$, and $N$ is the dimension of the state. This is the mean square error (MSE) between the original state and the output of the denoiser, which intends to reconstruct the original state. The temporal difference loss $\mathcal{L}_{\text{TD}}$ is defined as:

$$\mathcal{L}_{\text{TD}} = \begin{cases} \frac{1}{2\zeta}\eta^2, & \text{if } |\eta| < \zeta \\ |\eta| - \frac{\zeta}{2}, & \text{otherwise} \end{cases}, \quad \eta = r + \gamma \max_{a'} Q(s', a') - Q(D(\tilde{s}; \theta), a), \tag{4}$$

where $\zeta$ is set to 1. This is the Huber loss of the temporal difference, which is often used in DQN training. Note that the pretrained Q-network $Q$ can be replaced with robust agents such as Oikarinen et al. (2021) and Zhang et al. (2020) and our DS-DQN framework can also be combined with adversarial training to further improve the robustness. We will discuss this later in Section 5. The full training algorithm can be found in Appendix A.6.1 Algorithm 1.

**Issues of the smoothing strategy in CROP (Wu et al., 2022).** In the testing stage, we need to obtain the smoothed version of DRL agents. A simple way is to take the average of the output samples, which is the smoothing strategy used in CROP. However, this might not lead to a precise estimation of the certified radius since it requires estimating the output range $[V_{\min}, V_{\max}]$ of the Q-network. The certified radius proposed in CROP is shown as follows:

$$R_t = \frac{\sigma}{2}(\Phi^{-1}(\frac{\widetilde{Q}_{\text{CROP}}(s_t, a_1) - \Delta - V_{\min}}{V_{\max} - V_{\min}}) - \Phi^{-1}(\frac{\widetilde{Q}_{\text{CROP}}(s_t, a_2) + \Delta - V_{\min}}{V_{\max} - V_{\min}})), \quad (5)$$

where $R_t$ is the certified radius at time step $t$, $Q_{\text{CROP}} : \mathcal{S} \times \mathcal{A} \to [V_{\min}, V_{\max}]$, $\widetilde{Q}_{\text{CROP}}(s, a) = \frac{1}{m}\Sigma_{i=1}^{m}Q_{\text{CROP}}(s + \delta_i, a)$, $\delta_i \sim \mathcal{N}(0, \sigma^2 I_N), \forall i \in \{1, ..., m\}$, $a_1$ is the action with the largest Q-value among all the other actions, $a_2$ is the "runner-up" action, $\Delta = (V_{\max} - V_{\min})\sqrt{\frac{1}{2m}\ln\frac{1}{\alpha}}$, $\Phi$ is the CDF of standard normal distribution, $m$ is the number of the samples, and $\alpha$ is the one-side confidence parameter. Based on this expression, the output range of the Q-network $[V_{\min}, V_{\max}]$ can significantly affect the certified radius. The certified radius is small when the output range of the Q-network $[V_{\min}, V_{\max}]$ is large (e.g. Suppose $\widetilde{Q}_{\text{CROP}}(s_t, a_1) = 3$, $\widetilde{Q}_{\text{CROP}}(s_t, a_2) = -3$, $\sigma = 0.1$, $m = 100$, and $\alpha = 0.05$. Even though there is a significant gap between the two Q-values, the certified radius is only 0.007 under $[V_{\min}, V_{\max}] = [-10, 10]$. Instead, if we narrow down the interval to $[V_{\min}, V_{\max}] = [-3.5, 3.5]$, the certified radius grows to 0.086). CROP estimated $[V_{\min}, V_{\max}]$ by sampling some trajectories and finding the maximum and the minimum of the Q-values. However, if the actual interval is much larger than the estimation (which is likely to happen in practice since it is impossible to go over all the states), the calculated certified radius can be significantly overestimated.

**Our hard randomized smoothing strategy for testing.** To avoid the above issues, we leverage the hard Randomized Smoothing (hard RS) strategy to compute the certified radius without knowing $[V_{\min}, V_{\max}]$. We first define the hard Q-value $Q_h$ as follows:

$$Q_h(s, a) = \mathbb{1}_{\{a = \arg\max_{a'} Q(s, a')\}} \quad (6)$$

The output range of the hard Q-value $Q_h$ is always $[0, 1]$ and therefore does not lead to the aforementioned problem. Then, we define the hard RS for DS-DQN as follows:

$$\widetilde{Q}(s, a) = \mathbb{E}_{\delta \sim \mathcal{N}(0, \sigma^2 I_N)}Q_h(D(s + \delta), a). \quad (7)$$

We use Monte Carlo sampling to estimate $\widetilde{Q}$. The flow chart of the testing process of DS-DQN is shown in Appendix A.2 Figure 4 (b), and the full algorithm is in Appendix A.6.2 Algorithm 2.

**Our stronger attack.** The flow chart of attacking DS-DQN is shown in Appendix A.2 Figure 4 (c). Note that the policy of our DS-DQN $\tilde{\pi}(s) = \arg\max_a \widetilde{Q}(s, a)$ is a smoothed policy different from the base policy $\pi(s) = \arg\max_a Q(s, a)$. In CROP (Wu et al., 2022), they evaluated all the smoothed agents with the classic Projected Gradient Descent (PGD) attack. However, we found that the classic PGD attack is not effective in decreasing the reward of the smoothed agents as shown in Table 1. Hence, we introduce a new attack designed for the smoothed agents and evaluate the performance of DS-DQN based on this attack. The objective of our attack is to solve the below optimization problem:

$$\min_{\Delta s} \log \frac{\exp Q(D(\tilde{s} + \Delta s), a^*)}{\Sigma_a \exp Q(D(\tilde{s} + \Delta s), a)}, \text{ s.t. } ||\Delta s||_p \le \epsilon, \quad (8)$$

where $a^* = \arg\max_a \widetilde{Q}(s, a)$, $\widetilde{Q}(s, a)$ is defined in Eq.(7), $\tilde{s} = s + \mathcal{N}(0, \sigma^2 I_N)$, $\epsilon$ is the attack budget, and $p = 2$ or $\infty$ in our setting. Eq.(8) can be solved by PGD. In our new attack, the state with perturbation is added with a noise sampled from Gaussian distribution with the corresponding smoothing variance $\sigma$. We argue that this threat model is stronger than the classic PGD attack

Table 1: The comparison between our new attack and the classic PGD attack. Our attack reduces $51\%$ of the reward of DS-DQN on average, which is over $4.6\times$ stronger than $11\%$ of the classic PGD attack. We set the attack budget $\epsilon = 0.05$ in the $\ell_\infty$ attack, and $\epsilon = 0.9$ in the $\ell_2$ attack.

| Agents | Environments | No Attack | new $\ell_\infty$ Attack (Ours) | classic PGD $\ell_\infty$ Attack | new $\ell_2$ Attack (Ours) | classic PGD $\ell_2$ Attack |
|---|---|---|---|---|---|---|
| DS-DQN | Pong | $21.0 \pm 0.00$ | $\mathbf{18.8 \pm 1.17}$ | $19.4 \pm 2.33$ | $\mathbf{-16.8 \pm 3.70}$ | $20.8 \pm 0.40$ |
| | Freeway | $33.2 \pm 0.40$ | $\mathbf{23.8 \pm 1.17}$ | $30.4 \pm 1.85$ | $\mathbf{26.8 \pm 1.17}$ | $32.2 \pm 1.17$ |
| | RoadRunner | $35420 \pm 5116$ | $\mathbf{0 \pm 0}$ | $16200 \pm 1482$ | $\mathbf{13780 \pm 3396}$ | $37000.0 \pm 4565$ |

because now the attacker has the information of the smoothing variance $\sigma$. In practice, the attacker might not know the exact value of $\sigma$ and can only perform the classic PGD attack, which will be significantly weaker than our attack. The comparison of our attack against the classic PGD attack is in Table 1. The full algorithm of our attack is in Appendix A.6.3 Algorithm 3.

## 3.2 AS-PPO (ADVERSARIAL SMOOTHED - PROXIMAL POLICY OPTIMIZATION)

**Motivating example.** Unlike Atari games, the states of the continuous control tasks are often not image-based observations (e.g. Mujoco environment). Although we intended to make our algorithm designed for DQN also apply to PPO, we found that the Denoised Smoothing strategy did not work well in these environments. In particular, we found that the PPO agents are much more tolerant to the Gaussian noise and allow us to directly train the agent with RS. This inspires us to develop a Smoothed-PPO (S-PPO) agent, which is trained with RS. Table 3 row (c) in Section 5 shows the performance of the S-PPO agent. Although the S-PPO agent is more robust than the vanilla PPO agent, it is still not robust enough against the strongest attack. To resolve this issue, we propose AS-PPO based on adversarial training to enhance the robustness of our smoothed PPO agents. We describe the details of training, testing, and evaluating our AS-PPO in the following paragraphs.

**Training and loss function.** It is more complicated to do adversarial training in the RL setting than in the classification problem. Before we define the objective, we first define the smoothed policy $\tilde{\pi}$ of AS-PPO. We use the Median Smoothing (Chiang et al., 2020) strategy to smooth our agents. The median value has a nice property: it is almost unaffected by the outliers. Hence, Median Smoothing can give a better estimation of the expectation than mean smoothing when the number of samples is small. The smoothed version of AS-PPO is defined as follows:

$$\tilde{\pi}_i(a|s) = \mathcal{N}(\widetilde{M_i}, \widetilde{\Sigma}_i), \ \forall i \in \{1, ..., N_\text{action}\} \tag{9}$$

where $\widetilde{M_i} = \sup\{M \in \mathbb{R} | \mathbb{P}_{\delta \sim \mathcal{N}(0, \sigma^2 I_N)}[a_i^\text{mean} \leq M] \leq p\}$, $\widetilde{\Sigma}_i = \sup\{\Sigma \in \mathbb{R} | \mathbb{P}_{\delta \sim \mathcal{N}(0, \sigma^2 I_N)}[a_i^\text{std} \leq \Sigma] \leq p\}$, $(a_i^\text{mean}, a_i^\text{std})$ is the output of policy network given a state with noise $s + \delta$ as input, which represents the mean and standard deviation of the $i$-th coordinate of the action, $N_\text{action}$ is the dimension of the action, and $p$ is the percentile. This is the definition of the policy with Median Smoothing. Now, we can define the optimization problem as follows: $\max_\theta \min_{\{\Delta s_i\}_{i=1}^T} \mathbb{E}_t[\min(\frac{\tilde{\pi}(a_t|s_t + \Delta s_t; \theta)}{\tilde{\pi}(a_t|s_t + \Delta s_t; \theta_\text{old})} \hat{A}_t, \text{clip}(\frac{\tilde{\pi}(a_t|s_t + \Delta s_t; \theta)}{\tilde{\pi}(a_t|s_t + \Delta s_t; \theta_\text{old})}, 1 - \epsilon_\text{clip}, 1 + \epsilon_\text{clip})\hat{A}_t)]$, where $||\Delta s_t||_\infty \leq \epsilon$, $\hat{A}_t$ is the advantage, and $\epsilon_\text{clip}$ is the clipping hyperparameter. This is the objective of the smoothed PPO algorithm but with an inner min. By Danskin's theorem, we can first solve the inner minimization problem and then solve the outer maximization. This can be done by jointly training a policy network and an adversarial network. The adversary is another smoothed agent that is able to perturb the state and aims to minimize the surrogate reward. The smoothed adversarial policy is defined as follows:

$$\tilde{\mathcal{A}}_i(\Delta s|s) = \mathcal{N}(\widetilde{M_i}, \widetilde{\Sigma}_i), \ \forall i \in \{1, ..., N_\text{state}\} \tag{10}$$

where $\mathcal{A}$ is the adversary, $\widetilde{M_i} = \sup\{M \in \mathbb{R} | \mathbb{P}_{\delta \sim \mathcal{N}(0, \sigma^2 I_N)}[\Delta s_i^\text{mean} \leq M] \leq p\}$, $\widetilde{\Sigma}_i = \sup\{\Sigma \in \mathbb{R} | \mathbb{P}_{\delta \sim \mathcal{N}(0, \sigma^2 I_N)}[\Delta s_i^\text{std} \leq \Sigma] \leq p\}$, $(\Delta s_i^\text{mean}, \Delta s_i^\text{std})$ is the output of the adversarial network given a state with noise $s + \delta$ as input, which represents the mean and standard deviation of the $i$-th coordinate of the perturbation, $N_\text{state}$ is the dimension of the state, and $p$ is the percentile.

The flow chart of the training process is shown in Appendix A.3 Figure 5. In the policy update, we first collect the trajectories (with $50\%$ of the states being perturbed by the adversary) with the smoothed policy, and then update the value network and the policy network. In the adversary update, we collect the trajectories with the states always being perturbed, and then update the value network and the adversarial network. The full algorithm is in Appendix A.7.1 Algorithm 4 and 5.

**Testing.** We also use Median Smoothing in the testing to obtain the smoothed policy. However, we use the smoothed deterministic policy $\tilde{\pi}_{i,\text{det}}(s) = \widetilde{M}_i$, $\forall i \in \{1, ..., N_{\text{action}}\}$, where $\widetilde{M}_i = \sup\{M \in \mathbb{R} | \mathbb{P}_{\delta \sim \mathcal{N}(0, \sigma^2 I_N)}[a_i^{\text{mean}} \leq M] \leq p\}$, and $a_i^{\text{mean}}$ is the output of policy network given a state with noise $s + \delta$ as input ($a_i^{\text{mean}} = \pi_{i,\text{det}}(s + \delta)$) representing the mean of the $i$-th coordinate of the action. Here we only use the $a^{\text{mean}}$ value of the output of the policy network for smoothing.

**Attack.** To evaluate the performance of our AS-PPO, we modify several attack methods proposed in Zhang et al. (2020). They proposed the following attacks to evaluate the robustness of PPO agents: Random Attack, Critic Attack, Maximal Action Difference (MAD) Attack, and Minimum Robust Sarsa (Min-RS) Attack. More details about these attack algorithms can be found in Zhang et al. (2020). We also evaluate our AS-PPO under the Optimal Attack (Zhang et al., 2021), which is the current strongest attack for PPO using an adversarial agent to perturb the states. Our robustness evaluations for PPO are mainly based on these methods. However, the difference is that when we do PGD, the perturbed state is added with a noise sampled from Gaussian distribution with the smoothing variance $\sigma$. This setting is similar to the stronger attack we used while attacking the smoothed DQN agents.

## 4 ROBUSTNESS CERTIFICATION

The strength of the smoothed agents is that they come with certifiable robustness. Here we formally formulate the certified radius, action bound, and reward lower bound of our agents.

**Certified radius for DS-DQN.** The certified radius for our DS-DQN is defined as follows:

$$R_t = \frac{\sigma}{2}(\Phi^{-1}(\widetilde{Q}(s_t, a_1)) - \Phi^{-1}(\widetilde{Q}(s_t, a_2))), \tag{11}$$

where $a_1$ is the action with the largest Q-value among all the other actions, $a_2$ is the "runner-up" action, $R_t$ is the certified radius at time $t$, $\Phi$ is the CDF of normal distribution, $\sigma$ is the smoothing variance, and $\widetilde{Q}(s, a)$ is defined in Eq.(7). As long as the $\ell_2$ perturbation is bounded by $R_t$, the action will not change. Note that our expression of the certified radius is different from Eq.(5) proposed in CROP (Wu et al., 2022) since we use hard RS. The proof of the certified radius can be found in Appendix A.10.

**Action bound for AS-PPO.** Unfortunately, unlike the discrete action setting, there is no guarantee that the action will not change under a certain radius in the continuous action setting. Hence, we propose the **Action Bound**, which bounds the policy of smoothed PPO agents in a close region:

$$\tilde{\pi}_{\text{det},\underline{p}}(s_t) \preceq \tilde{\pi}_{\text{det},p}(s_t + \Delta s) \preceq \tilde{\pi}_{\text{det},\overline{p}}(s_t), \text{ s.t. } ||\Delta s||_2 \leq \epsilon, \tag{12}$$

where $\tilde{\pi}_{i,\text{det},p}(s) = \sup\{a_i \in \mathbb{R} | \mathbb{P}_{\delta \sim \mathcal{N}(0, \sigma^2 I_N)}[\pi_{i,\text{det}}(s + \delta) \leq a_i] \leq p\}, \forall i \in \{1, ..., N_{\text{action}}\}$, $\underline{p} = \Phi(\Phi^{-1}(p) - \frac{\epsilon}{\sigma})$, $\overline{p} = \Phi(\Phi^{-1}(p) + \frac{\epsilon}{\sigma})$, and $p$ is the percentile. The proof of the action bound can be found in Appendix A.11. We designed a metric based on this action bound to evaluate the certified robustness for smoothed PPO agents. See Appendix A.13 for more details.

**Reward lower bound for smoothed agents.** By viewing the whole trajectory as a function $F_\pi$, we define $F_\pi : \mathbb{R}^{H \times N} \to \mathbb{R}$ that maps the vector of perturbations for the whole trajectory $\Delta s = [\Delta s_0, ..., \Delta s_{H-1}]^T$ to the cumulative reward. Then, the reward lower bound is defined as follows:

$$\widetilde{F}_{\pi,p}(\Delta s) \geq \widetilde{F}_{\pi,\underline{p}}(0), \text{ s.t. } ||\Delta s||_2 \leq B, \tag{13}$$

where $\widetilde{F}_{\pi,p}(\Delta s) = \sup\{r \in \mathbb{R} | \mathbb{P}_{\delta \sim \mathcal{N}(0, \sigma^2 I_{H \times N})}[F_\pi(\delta + \Delta s) \leq r] \leq p\}$, $\widetilde{F}_{\pi,\underline{p}}(0) = \sup\{r \in \mathbb{R} | \mathbb{P}_{\delta \sim \mathcal{N}(0, \sigma^2 I_{H \times N})}[F_\pi(\delta) \leq r] \leq \underline{p}\}$, $\delta = [\delta_0, ..., \delta_{H-1}]^T$, $\underline{p} = \Phi(\Phi^{-1}(p) - \frac{B}{\sigma})$, $H$ is the length of the trajectory, and $B$ is the $\ell_2$ attack budget for the entire trajectory. If the attack budget of each state is $\epsilon$, then $B = \epsilon\sqrt{H}$. This bound ensures that the reward will not fall below a certain value while given any $\ell_2$ perturbation with budget $B$. The proof of the reward lower bound can be found in Appendix A.12.

In practice, we use Monte Carlo sampling to estimate all the bounds in Section 4, and hence, it is necessary to introduce the confidence interval which can change the bounds while the sample number is different. We give the detailed formula of the bounds we used to conduct all the experiments in Appendix A.9.

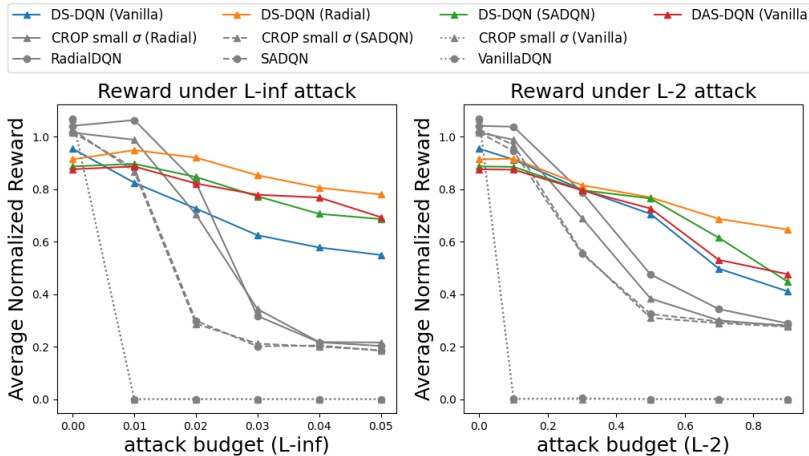

Figure 2: The average normalized reward of different DQN agents. Our agents achieve high robust reward under attack with a large budget.

## 5 EXPERIMENT

**Setup.** In our DQN settings, the evaluations are done in three Atari environments — Pong, Freeway, and RoadRunner. We train the denoiser $D$ with different base models and with adversarial training. Our methods are listed as follows:

- DS-DQN (Vanilla): using simple DQN as pretrained Q-network.
- DS-DQN (Radial) and DS-DQN (SADQN): using RadialDQN (Oikarinen et al., 2021) and SADQN (Zhang et al., 2020) as pretrained Q-network respectively.
- DAS-DQN (Denoised Adversarial Smoothed - DQN) (Vanilla): the implementation of DS-DQN (Vanilla) combined with adversarial training.

We compare our DS-DQN with the following baselines:

- RadialDQN (Oikarinen et al., 2021): the current state-of-the-art robust agent.
- SADQN (Zhang et al., 2020): a robust agent.
- CROP (Wu et al., 2022): the smoothed agents. CROP small $\sigma$ are the implementation of the CROP framework using a relatively small smoothing factor.

In our PPO settings, the evaluations are done on three continuous control tasks in the Mujoco environments — Walker, Hopper, and Humanoid. We train each agent 15 times and report the median performance as suggested in Zhang et al. (2020) for a fair comparison. We compare our AS-PPO with the following baselines:

- SAPPO (SGLD) and (Convex) (Zhang et al., 2020): the two SAPPO implementations.
- SAPPO+RS (SGLD) and (Convex): the naively smoothed SAPPO agents similar to CROP.

See Appendix A.8 for more details about our setting.

**Evaluation of DS-DQN.** The robust reward under $\ell_\infty$ and $\ell_2$ PGD attack of our DS-DQN is shown in Figure 2. Note that we use our stronger attack setting introduced in Section 3.1 to evaluate all the smoothed agents. Our DS-DQNs and DAS-DQN receive higher reward than the current state-of-the-art RadialDQN under a large attack budget. It can be seen that our DS-DQN (Vanilla) is already more robust than RadialDQN even without further using other robust agents as base models. RadialDQN and CROP small $\sigma$ (Radial) achieve similar performance indicating that the CROP framework cannot improve the empirical robustness if using small $\sigma$ to avoid decreasing the clean reward. More detailed experiment results and discussion about the robust reward under attack can be found in Appendix A.15. For the robustness certification, our methods exhibit large certified radius and high reward lower bound without compromising the clean reward, which is shown in Table 2. More detailed experiment results of the reward lower bound can be found in Appendix A.14.

Table 2: The average clean reward, certified radius, and reward lower bound of different smoothed agents. Our agents have much better clean reward and robustness. The agents with extremely low reward are highlighted in red. The reward lower bound is calculated while given any $\ell_2$ perturbation with budget $\epsilon = 0.005$ at each state.

| Environments | Average Certified Radius ↑ | | | Average Clean Reward ↑ | | | Reward Lower Bound ↑ | | |
|---|---|---|---|---|---|---|---|---|---|
| | Pong | Freeway | RoadRunner | Pong | Freeway | RoadRunner | Pong | Freeway | RoadRunner |
| **Our proposed methods:** | | | | | | | | | |
| DS-DQN (Vanilla) | 0.0614 | 0.0891 | 0.0739 | **21.0** | 33.2 | 35420 | 12.0 | **26.0** | 16500 |
| DS-DQN (Radial) | 0.1046 | 0.1316 | 0.0962 | **21.0** | 32.2 | 31760 | 19.0 | 25.0 | 12921 |
| DS-DQN (SADQN) | 0.1157 | 0.1362 | 0.0886 | **21.0** | 30.0 | 31080 | **19.7** | 24.0 | **19170** |
| DAS-DQN (Vanilla) | 0.0509 | 0.0970 | 0.0667 | 18.8 | 32.6 | 28840 | 11.0 | 26.0 | 17770 |
| **Baselines:** | | | | | | | | | |
| **(a) CROP** | | | | | | | | | |
| CROP (Vanilla) | 0.0027 | 0.0432 | 0.0078 | −20.4 | 22.2 | 17060 | −21.0 | 8.0 | 0 |
| CROP (Radial) | **0.1163** | **0.1396** | 0.1084 | −21.0 | 22.2 | 9180 | −21.0 | 19.0 | 3000 |
| CROP (SADQN) | 0.1152 | **0.1396** | 0.1025 | −21.0 | 22.2 | 14060 | −21.0 | 19.0 | 3300 |
| **(b) Naive training with RS** | | | | | | | | | |
| S-DQN | **0.1163** | 0.0998 | **0.1163** | −21.0 | 0.0 | 960 | −21.0 | 0.0 | 900 |

**Evaluation of AS-PPO.** The clean reward, reward lower bound, and robust reward under Min-RS attack of our AS-PPO is shown in Table 3. Note that we also use our stronger attack setting introduced in Section 3.2 to evaluate all the smoothed PPO agents. Our AS-PPO has the highest reward under the Min-RS attack and also exhibits a better trade-off between the clean reward and robustness certification (the reward lower bound). Through comparing rows (a) and (b), the clean reward is degraded and the reward under attack only slightly improves when RS is present, which suggests that naively applying RS similar to the CROP (Wu et al., 2022) framework during the testing cannot address the issue of the poor trade-off. In addition, our AS-PPO receives a much higher clean reward on average, which shows that the randomized smoothing approach can further help boost performance in the non-adversarial setting. The **Optimal Attack** results is shown in Table 4. We directly compare to the SAPPO implementation in Zhang et al. (2021) to

Table 3: The average normalized reward of different PPO agents. Our agents achieve high clean reward, robust reward, and robustness guarantee at the same time. The reward lower bound is calculated while given any $\ell_2$ perturbation with budget $\epsilon = 0.01$ at each state. Note that the $\ell_2$ budget of reward lower bound is smaller than the $\ell_\infty$ budget of Min-RS attack. Hence, it is possible to have a higher lower bound than the reward under attack.

| Methods | Average Normalized Reward ↑ | | |
|---|---|---|---|
| | Clean Reward | Min-RS Attack | Lower Bound |
| **Our proposed methods:** | | | |
| AS-PPO | **0.997** | **0.720** | 0.624 |
| **Baselines:** | | | |
| **(a) Smoothed agents** | | | |
| SAPPO+RS (SGLD) | 0.903 | 0.634 | **0.639** |
| SAPPO+RS (Convex) | 0.943 | 0.514 | 0.598 |
| VanillaPPO+RS | 0.767 | 0.224 | 0.179 |
| **(b) Non-smoothed agents** | | | |
| SAPPO (SGLD) | 0.963 | 0.597 | — |
| SAPPO (Convex) | 0.940 | 0.482 | — |
| VanillaPPO | 0.847 | 0.181 | — |
| **(c) Naive training with RS** | | | |
| S-PPO | 0.849 | 0.491 | 0.527 |

ensure a fair comparison. Our AS-PPO also performs well under the optimal attack which suggests that our AS-PPO is still more robust under the state-of-the-art attack for PPO. More detailed experiment results and discussion about the robust reward under different attacks and the reward lower bound can be found in Appendix A.15 and A.14 respectively.

## 6 BACKGROUND AND RELATED WORKS

**Randomized Smoothing (RS).** Randomized Smoothing (Cohen et al., 2019) has been proved to provide robustness guarantee to a *smoothed* classifier under $\ell_2$ perturbation on input examples. The idea is to transform an arbitrary base classifier into an $L$-Lipschitz smoothed classifier by adding Gaussian noises to the input. This transformation facilitates *black-box* robustness verification on the smoothed classifier, which ensures the classification result remains unchanged within the certified radius without the need to know the model parameters. This can be formulated as below. Given a base classifier $f : \mathbb{R}^d \to \mathcal{Y}$, and let $\tilde{f} : \mathbb{R}^d \to \mathcal{Y}$ be the

Table 4: The Optimal Attack results of our AS-PPO and SAPPO performance reported in Zhang et al. (2021).

| Environment | Optimal Attack | |
|---|---|---|
| | Walker | Hopper |
| **Our proposed methods:** | | |
| AS-PPO | **4296** | **1500** |
| **Baseline:** | | |
| SAPPO in (Zhang et al., 2021) | 2908 | 1076 |

Table 5: The comparison between our proposed methods and other robust DRL agents. Our methods are desirable in both empirical robustness and robustness guarantee.

| Methods | Empirical Robustness | | Robustness Guarantee | | |
|---|---|---|---|---|---|
| | Clean Reward↑ | Reward under Attack↑ | Certified Radius (DQN) | Action bound (PPO) | Reward lower bound↑ (DQN and PPO) |
| **Our methods:** | | | | | |
| DS-DQN & AS-PPO | **High** | **Highest** | **Yes** | **Yes** | **High** |
| **Baselines:** | | | | | |
| SADQN & SAPPO | High | High | No | No | No |
| RADIAL-RL | High | High | No | No | No |
| CROP | Low | Low | Yes | No PPO implementation | Low |

smoothed classifier (i.e., $f$ after RS), $\tilde{f}$ can be expressed as $\tilde{f}(x) = \arg\max_{c \in \mathcal{Y}} \mathbb{P}_{\delta \sim \mathcal{N}(0,\sigma^2 I)}[f(x + \delta) = c]$, where $\delta$ is a random vector following Gaussian distribution $\mathcal{N}(0, \sigma^2 I)$. The smoothed classifier $\tilde{f}$ predicts class $c_A$ with probability $p_A$, and predicts the "runner-up" class $c_B$ with probability $p_B$. The certified radius of $\tilde{f}$ is denoted as $R$ such that $\tilde{f}(x + \Delta) = \tilde{f}(x), \ \forall ||\Delta||_2 \leq R$. $R$ can be derived as $R = \frac{\sigma}{2}(\Phi^{-1}(p_A) - \Phi^{-1}(p_B))$, where $\Phi^{-1}$ is the inversed Gaussian CDF. When we replace $p_A$ and $p_B$ by $\underline{p_A}$ and $\overline{p_B}$, where $\underline{p_A}$ is the lower confidence bound of $p_A$, and $\overline{p_B}$ is the upper confidence bound of $p_B$, the certified radius still holds. In practice, we can use Monte Carlo sampling to estimate $\underline{p_A}$ and $\overline{p_B}$.

**Denoised Smoothing (Salman et al., 2020).**   In Salman et al. (2020), the authors proposed to add a denoiser before the original image classifier with the goal of removing the Gaussian noises introduced by RS. This approach gives the classifier the ability to tolerate large noises. Our method is the first work leveraging Denoised Smoothing in the DRL setting.

**Training robust DRL agents.**   There are several existing works of learning robust DRL agents:

- **SADQN and SAPPO (Zhang et al., 2020):** To train a robust policy, Zhang et al. (2020) derived a robust regularizer based on the total variation distance and KL-divergence between the perturbed policies and the original policies. They proposed SADQN for robust DQN and proposed SAPPO (SGLD) and SAPPO (Convex) for robust PPO.

- **RADIAL-RL (Oikarinen et al., 2021):** RadialRL is the state-of-the-art DRL agent against $\ell_p$-norm attack on both Atari games and continuous control tasks. The key idea of their approach is to use the adversarial loss as a regularizer based on the robustness verification bounds.

- **CROP (Wu et al., 2022):** CROP is the first framework using RS to study the robustness certification of DRL agents. However, they only transform existing pretrained DRL agents into smoothed agents by exploiting RS at the testing stage. Their approach exhibits a significant trade-off between the clean reward and certified radius as we discussed in Section 2. This failure case shows that the CROP agents are not usable in practice because robust but poorly performed agents are not useful. Therefore, it is necessary to apply our proposed method.

The comparison between our method and the above robust DRL agents is shown in Table 5.

# 7   CONCLUSION AND FUTURE WORKS

In this work, we have shown with extensive experiments that our proposed DS-DQN and AS-PPO agents can mitigate the trade-off between robustness and clean reward, unlike the CROP agents. Our agents achieve high clean reward and are robust in terms of both robustness certificates and robust reward against the current strongest attack, establishing the new state-of-the-art in the field. In future work, we are planning to investigate the idea of leveraging robustness certificates into training to further strengthen the robustness and utility of DRL agents.

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

# A  APPENDIX

**Contents**

### A.1 OVERVIEW OF OUR FRAMEWORK

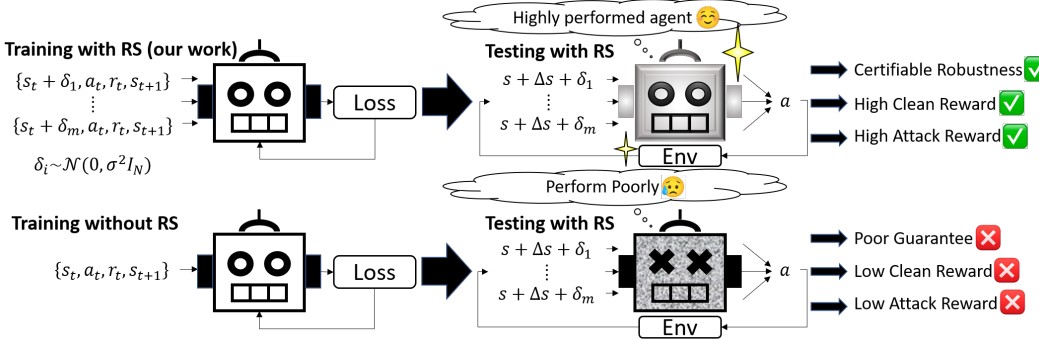

Figure 3: The overview of our framework. Simply applying randomized smoothing at the testing stage fails in robustness guarantee as well as defending against attacks.

### A.2 THE PIPELINE OF OUR DS-DQN

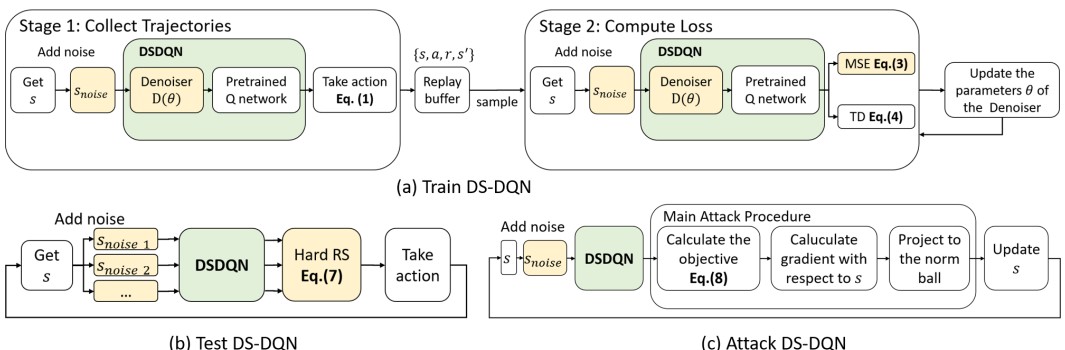

Figure 4: The flow chart of: (a) training process of DS-DQN, (b) testing process of DS-DQN, (c) our new attack pipeline for DS-DQN, which can effectively decrease the reward of any smoothed agents.

### A.3 THE PIPELINE OF OUR AS-PPO

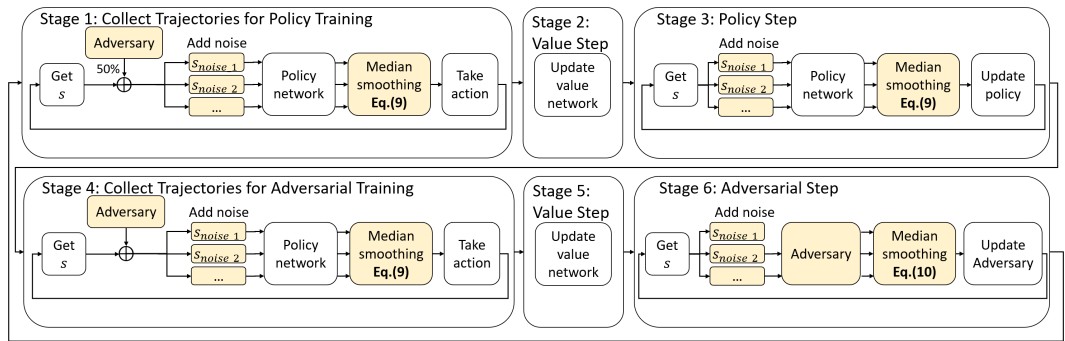

Figure 5: The flow chart of the training process of AS-PPO.

### A.4 DETAILED CONTRIBUTIONS OF OUR WORK

- Contributions related to the CROP (Wu et al., 2022) framework:
  - We identify the failure mode of the existing smoothed DRL agents, showing that CROP agents have poor trade-offs on the clean reward and robustness (see Section 2).
  - We point out that the robustness of the CROP agents might be overestimated due to the smoothing strategy and attack they used. We fix this issue by introducing hard RS and a stronger attack.
  - We extend their proposed robust guarantee for DQN agents to the PPO setting and defined action bound. To our best knowledge, the action bound for PPO has never been derived before. We also do experiments on this bound (see Appendix A.13).
- Contributions related to robust DRL agents:
  - Our agents achieved the state-of-the-art results under attack. In the discrete action setting, our DS-DQN earns $2.07\times$ more reward on average than the current best agent under the strongest attack. In the continuous action setting, our AS-PPO earns $1.25\times$ more reward under the strongest attack.
  - Our agent is the first state-of-the-art agents with high robustness guarantee at the same time (certified radius and reward lower bound), while the previous state-of-the-art only evaluate their agents under empirical attack.
- Contributions related to Randomized Smoothing (RS) in DRL:
  - We develop the first robust DRL training algorithms leveraging randomized smoothing for both discrete actions (DS-DQN) and continuous actions (AS-PPO).
  - We show that simply training with RS does not work, and is necessary to use denoised smoothing and adversarial training.
  - We point out that different smoothing strategies can affect the robustness guarantee. (i.e. Our hard RS strategy is not affected by the output range of the Q-network and generally achieves larger certified radius.)
- Contributions related to adversarial attack:
  - We develop a new attack aiming at attacking smoothed agents. This attack model we proposed can be easily used on any attack based on optimization.

## A.5 Comparison of our DS-DQN with CROP using small smoothing factor

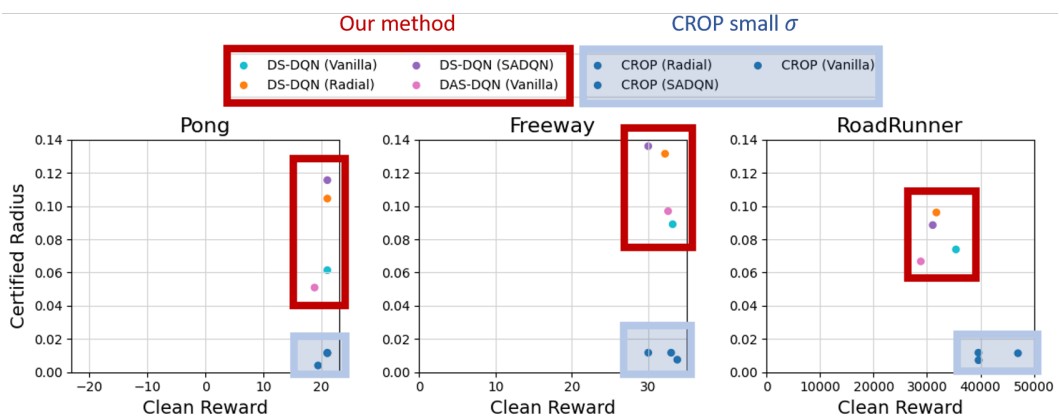

Figure 6: The average clean reward and certified radius of our DS-DQN and CROP (small $\sigma$). Our agents have much larger certified radius compared to the CROP framework with small $\sigma$.

## A.6 Detailed algorithms of DS-DQN

### A.6.1 Training algorithm of DS-DQN

The training algorithm of DS-DQN is shown in Algorithm 1. The algorithm includes all the details of the training procedure introduced in Section 3.1. We first add a noise to the current state and take action with $\epsilon$-greedy strategy, Then, store the transitions $\{s_t, a_t, r_t, s_{t+1}\}$ into the replay buffer. Note that the state $s_t$ we stored here is the clean state without noise. When updating the denoiser $D$, we sample a batch of transitions from the replay buffer, add noise to the state again, and compute the loss.

---

**Algorithm 1** Train DS-DQN

---

1: **Input:** smoothing variance $\sigma$, steps $T$, replay buffer $\mathcal{B}$, Denoiser $D$, pretrained Q network $Q$
2: **for** $t = 1$ **to** $T$ **do**
3:  Sample a noise from the normal distribution and add to the state $\tilde{s}_t = s_t + \mathcal{N}(0, \sigma^2 I_N)$
4:  Select a random action $a_t$ with probability $\epsilon_t$, otherwise $a_t = \arg\max_a Q(D(\tilde{s}_t; \theta), a)$
5:  Store the transition $\{s_t, a_t, r_t, s_{t+1}\}$ in $\mathcal{B}$
6:  Sample a batch of samples $\{s, a, r, s'\}$ from $\mathcal{B}$
7:  Sample a noise from the normal distribution and add to the state $\tilde{s} = s + \mathcal{N}(0, \sigma^2 I_N)$
8:  Compute the reconstruction loss $\mathcal{L}_{\text{R}} = \text{MSE}(D(\tilde{s}; \theta), s)$
9:  Compute the temporal difference loss $\mathcal{L}_{\text{TD}} = \text{Huber}(r + \gamma \max_{a'} Q(s', a') - Q(D(\tilde{s}; \theta), a))$
10:  Total loss $\mathcal{L} = \lambda_1 \mathcal{L}_{\text{R}} + \lambda_2 \mathcal{L}_{\text{TD}}$
11:  Perform gradient descent to minimize loss $\mathcal{L}$ and update the parameters $\theta$ of the denoiser $D$
12: **end for**

---

### A.6.2 Testing algorithm of DS-DQN

The testing algorithm of DS-DQN is shown in Algorithm 2. The algorithm includes all the details of the testing procedure introduced in Section 3.1. We use the hard Randomized Smoothing strategy to smooth our agent and do Monte Carlo sampling to estimate the expectation. The definition of $Q_h$ is in Eq.(6).

---

**Algorithm 2** Test DS-DQN

---

1: **Input:** smoothing variance $\sigma$, number of samples $M$, number of the actions $N$, Denoiser $D$, pretrained Q network $Q$
2: **while** not end game **do**
3:     Get state $s$ from the environment
4:     **for** $m = 1$ **to** $M$ **do**
5:         Sample a noise from the normal distribution and add to the state $\tilde{s}_m = s_m + \mathcal{N}(0, \sigma^2 I_N)$
6:         Store the $Q_h$ value of all the actions $[Q_h(D(\tilde{s}_m), a_1), ..., Q_h(D(\tilde{s}_m), a_N)]$ to the list
7:     **end for**
8:     Take the mean of the $Q_h$ value of each action $\widetilde{Q}(s, a_n) = \frac{1}{M} \Sigma_{m=1}^{M} Q_h(D(\tilde{s}_m), a_n)$
9:     Choose the action with the maximum $\widetilde{Q}$ value $a^* = \arg\max_{a_n} \widetilde{Q}(s, a_n)$
10:    Take action and get the reward
11: **end while**
12: Return the total reward

---

### A.6.3 ATTACK ALGORITHM OF DS-DQN

The algorithm of attacking DS-DQN is shown in Algorithm 3. The algorithm includes all the details of the attack procedure introduced in Section 3.1. Note that our new attack considers the noise caused by randomized smoothing while doing PGD.

---

**Algorithm 3** New PGD attack designed for DS-DQN

---

1: **Input:** number of iterations $T$, attack budget $\epsilon$, smoothing variance $\sigma$, number of samples $M$, Denoiser $D$, pretrained Q network $Q$
2: Get state $s$ from the environment
3: $\hat{s} = s$
4: **for** $t = 1$ **to** $T$ **do**
5:     Sample a noise from the normal distribution and add to the state $\tilde{\hat{s}} = \hat{s} + \mathcal{N}(0, \sigma^2 I_N)$
6:     Compute the cross-entropy loss
    $\mathcal{L} = -\log \frac{\exp(Q(D(\tilde{\hat{s}}), a^*))}{\Sigma_a \exp(Q(D(\tilde{\hat{s}}), a))}$,
    where $a^*$ is the original optimal action decided by the agent
7:     Calculate the gradient with respect to $\hat{s}$, and project to the $\ell_2$ or $\ell_\infty$ norm ball
8:     Update $\hat{s}$ by adding the gradient
9: **end for**
10: Return the perturbed state $\hat{s}$

---

### A.7 Detailed algorithms of AS-PPO

#### A.7.1 Training algorithm of AS-PPO

The training algorithm of AS-PPO is shown in Algorithm 4 and 5. The algorithm includes all the details of the training procedure introduced in Section 3.2. The algorithm of CollectTrajectories function used in the step A1 and B1 of Algorithm 4 is shown in Algorithm 5. Note that in step A1, there is a 50% chance to add a perturbation to the state. However, when we do step B1, we always use the perturbed state.

---

**Algorithm 4** Train AS-PPO

---

1: **Input:** smoothing variance $\sigma$, attack budget $\epsilon$, number of samples $M$, iterations $T$, Policy network $\pi$, Value network $V$, adversary network $\mathcal{A}$
2: **for** $t = 1$ **to** $T$ **do**
3:     **// Step A1: Collect trajectories for policy training**
    $\{\tau_k\} = $ **CollectTrajectories(CollectAdversaryTrajectories = False)**
4:     Compute cumulative reward $\hat{R}_{k,i}$ for each step $i$ in episode $k$ with discount factor $\gamma$
5:     **// Step A2: Update the value network with loss**
    $\mathcal{L}_V(\theta) = \frac{1}{\Sigma_k |\tau_k|} \Sigma_{\tau_k} \Sigma_i (V(s_{k,i}) - \hat{R}_{k,i})^2$
6:     **// Step A3: Update the policy network**
7:     **for** $m = 1$ **to** $M$ **do**
8:         Sample a noise from the normal distribution and add to the state $\tilde{s}_{k,i,m} = s_{k,i,m} + \mathcal{N}(0, \sigma^2 I_N)$
9:         Store the output of the policy network $(a_{k,i,m}^{\text{mean}}, a_{k,i,m}^{\text{std}})$ to the list, where $\mathcal{N}(a_{k,i,m}^{\text{mean}}, a_{k,i,m}^{\text{std}}) = \pi(a_{k,i,m}|\tilde{s}_{k,i,m})$
10:     **end for**
11:     Take the median and obtain the smoothed policy
    $\tilde{\pi}(a_{k,i}|s_{k,i}) = \mathcal{N}(\text{median}(a_{k,i,1}^{\text{mean}}, ..., a_{k,i,M}^{\text{mean}}), \text{median}(a_{k,i,1}^{\text{std}}, ..., a_{k,i,M}^{\text{std}}))$
12:     Update the policy network with the PPO loss
    $\mathcal{L}(\theta) = -\frac{1}{\Sigma_k |\tau_k|} \Sigma_{\tau_k} \Sigma_i \min(\frac{\tilde{\pi}(a_{k,i}|s_{k,i};\theta)}{\tilde{\pi}(a_{k,i}|s_{k,i};\theta_{\text{old}})} \hat{A}_{k,i}, \text{clip}(\frac{\tilde{\pi}(a_{k,i}|s_{k,i};\theta)}{\tilde{\pi}(a_{k,i}|s_{k,i};\theta_{\text{old}})}, 1 - \epsilon_{\text{clip}}, 1 + \epsilon_{\text{clip}})\hat{A}_{k,i})$,
    where $\hat{A}_{k,i}$ is the advantage
13:     **// Step B1: Collect trajectories for adversarial training**
    $\{\tau_k\} = $ **CollectTrajectories(CollectAdversaryTrajectories = True)**
14:     Compute cumulative reward $\hat{R}_{k,i}$ for each step $i$ in episode $k$ with discount factor $\gamma$
15:     **// Step B2: Update the value network again**
    $\mathcal{L}_V(\theta) = \frac{1}{\Sigma_k |\tau_k|} \Sigma_{\tau_k} \Sigma_i (V(s_{k,i}) - \hat{R}_{k,i})^2$
16:     **// Step B3: Update the adversarial network**
17:     Negate the reward for adversarial training
18:     **for** $m = 1$ **to** $M$ **do**
19:         Sample a noise from the normal distribution and add to the state $\tilde{s}_{k,i,m} = s_{k,i,m} + \mathcal{N}(0, \sigma^2 I_N)$
20:         Store the output of the adversarial network $(\Delta s_{k,i,m}^{\text{mean}}, \Delta s_{k,i,m}^{\text{std}})$ to the list, where $\mathcal{N}(\Delta s_{k,i,m}^{\text{mean}}, \Delta s_{k,i,m}^{\text{std}}) = \mathcal{A}(\Delta s_{k,i,m}|\tilde{s}_{k,i,m})$
21:     **end for**
22:     Take the median and obtain the smoothed adversary
    $\tilde{\mathcal{A}}(\Delta s_{k,i}|s_{k,i}) = \mathcal{N}(\text{median}(\Delta s_{k,i,1}^{\text{mean}}, ..., \Delta s_{k,i,M}^{\text{mean}}), \text{median}(\Delta s_{k,i,1}^{\text{std}}, ..., \Delta s_{k,i,M}^{\text{std}}))$
23:     Update the adversarial network by the adversary loss
    $L(\theta) = \frac{1}{\Sigma_k |\tau_k|} \Sigma_{\tau_k} \Sigma_i \min(\frac{\tilde{\mathcal{A}}(\Delta s_{k,i}|s_{k,i};\theta)}{\tilde{\mathcal{A}}(\Delta s_{k,i}|s_{k,i};\theta_{\text{old}})} \hat{A}_{k,i}, \text{clip}(\frac{\tilde{\mathcal{A}}(\Delta s_{k,i}|s_{k,i};\theta)}{\tilde{\mathcal{A}}(\Delta s_{k,i}|s_{k,i};\theta_{\text{old}})}, 1 - \epsilon_{\text{clip}}, 1 + \epsilon_{\text{clip}})\hat{A}_{k,i})$,
    where $\hat{A}_{k,i}$ is the advantage (calculated with the negated reward)
24: **end for**

---

---

**Algorithm 5 CollectTrajectories** function

---

1: **Input:** number of trajectories $K$, probability of using adversarial examples $p$, attack budget $\epsilon$, smoothing variance $\sigma$, number of samples $M$, Policy network $\pi$, adversary network $\mathcal{A}$
2: **for** $k = 1$ **to** $K$ **do**
3:    **while** not end game **do**
4:       Get state $s$ from the environment
5:       **if** CollectAdversaryTrajectories **or** random$(0, 1) \leq p$ **then**
6:          $\Delta s \sim \mathcal{A}(\Delta s|s)$
7:          $s = s + \epsilon \tanh(\Delta s)$
8:       **end if**
9:       **for** $m = 1$ **to** $M$ **do**
10:          Sample a noise from the normal distribution and add to the state $\tilde{s}_m = s_m + \mathcal{N}(0, \sigma^2 I_N)$
11:          Store the mean and standard deviation of the action $(a_m^{\mathrm{mean}}, a_m^{\mathrm{std}})$ to the list, where $\mathcal{N}(a_m^{\mathrm{mean}}, a_m^{\mathrm{std}}) = \pi(a|\tilde{s}_m)$
12:       **end for**
13:       Take the median and obtain the smoothed policy $\tilde{\pi}(a|s) = \mathcal{N}(\mathrm{median}(a_1^{\mathrm{mean}}, ..., a_M^{\mathrm{mean}}), \mathrm{median}(a_1^{\mathrm{std}}, ..., a_M^{\mathrm{std}}))$
14:       Take action with the smoothed policy and collect the reward
15:    **end while**
16:    Store the trajectory $\tau_k$
17: **end for**
18: Return the set of the trajectories $\{\tau_k\}$

---

## A.8 DETAILED SETTINGS FOR DQN AND PPO

### A.8.1 SETTINGS FOR DQN

Our DQN implementation is based on the SADQN (Zhang et al., 2020) and CROP (Wu et al., 2022). We use the pretrained VanillaDQN agent without any robust training as our base model and the DnCNN structure proposed in Zhang et al. (2017) as the denoiser to train DS-DQN (Vanilla). We train our DS-DQN for $300,000$ frames in every environment. The training time of DS-DQN is roughly 12 hours on our hardware, which is much faster than 40 hours of SADQN and 17 hours of RadialDQN. The smoothing variance $\sigma$ for all the agents other than CROP small $\sigma$ is set to 0.1 in Pong, 0.12 in Freeway, and 0.1 in RoadRunner. The smoothing variance of CROP small $\sigma$ is set to 0.01. All the experiment results under attack are obtained by taking the average of 5 episodes.

### A.8.2 SETTINGS FOR PPO

Our PPO implementation is based on the SAPPO (Zhang et al., 2020). We train AS-PPO for $2500 \times 2048$ steps in Walker and Hopper, and $8000 \times 2048$ steps in Humanoid. Only $50\%$ of the policy updates use adversary examples to ensure that AS-PPO also achieves high clean reward. Note that there is a variance between the performance of each agent trained with the same algorithm. To get a fair and comparable result, we train each agent 15 times and reported the median of the performance as suggested in Zhang et al. (2020). The smoothing variance $\sigma$ for all the smoothed agents is set to 0.2 in Walker, 0.3 in Hopper, and 0.4 in Humanoid. The $\ell_\infty$ attack budget for all the attacks for PPO (Random, Critic, MAD, Min-RS, and Optimal Attack) is set to 0.05 in Walker and 0.075 in Hopper and Humanoid. All the experiment results under attack are obtained by taking the average of 50 episodes.

## A.9 DETAILS OF ESTIMATING BOUNDS

### A.9.1 ESTIMATING THE CERTIFIED RADIUS FOR DS-DQN

In practice, we use Monte Carlo sampling to estimate $\widetilde{Q}$, which denotes as $\widetilde{Q}_{\text{est}}$. The estimation of the Certified Radius is formulated as follows:

$$R_{\text{est},t} = \frac{\sigma}{2}(\Phi^{-1}(\widetilde{Q}_{\text{est}}(s_t, a_1) - \Delta) - \Phi^{-1}(\widetilde{Q}_{\text{est}}(s_t, a_2) + \Delta)), \tag{14}$$

where $\widetilde{Q}_{\text{est}}(s, a) = \frac{1}{m}\Sigma_{i=1}^m Q_h(D(s + \delta_i), a), \delta_i \sim \mathcal{N}(0, \sigma^2 I_N), \forall i \in \{1, ..., m\}, \Delta = \sqrt{\frac{1}{2m}\ln\frac{1}{\alpha}}$, $m$ is the number of the samples ($m = 100$ in our setting), and $\alpha$ is the one-side confidence parameter ($\alpha = 0.05$ in our setting). The proof of this estimation can be found in Appendix A.10.

### A.9.2 ESTIMATING THE ACTION BOUND FOR AS-PPO

In practice, we use Monte Carlo sampling to estimate $\tilde{\pi}_{\text{det},p}$, which denotes as $\tilde{\pi}_{\text{det},p_{\text{est}}}$. The estimation of the Action Bound is formulated as follows:

$$\tilde{\pi}_{\text{det},\underline{p_{\text{est}}}}(s_t) \preceq \tilde{\pi}_{\text{det},p_{\text{est}}}(s_t + \Delta s) \preceq \tilde{\pi}_{\text{det},\overline{p_{\text{est}}}}(s_t), \ s.t \ ||\Delta s||_2 \leq \epsilon, \tag{15}$$

where $\tilde{\pi}_{i,\text{det},p_{\text{est}}}(s) = max\{a_i \in \mathbb{R}| \ |\{x \in S_i|x \leq a_i\}| \leq \lceil mp_{\text{est}}\rceil\}, S_i = \{\pi_{i,\text{det}}(s + \delta_1), ..., \pi_{i,\text{det}}(s + \delta_m)\}, \forall i \in \{1, ..., N_{\text{action}}\}, \delta_j \sim \mathcal{N}(0, \sigma^2 I_N), \forall j \in \{1, ..., m\}, \underline{p_{\text{est}}} = \Phi(\Phi^{-1}(p_{\text{est}} - \Delta) - \frac{\epsilon}{\sigma}), \overline{p_{\text{est}}} = \Phi(\Phi^{-1}(p_{\text{est}} + \Delta) + \frac{\epsilon}{\sigma}), \Delta = \sqrt{\frac{1}{2m}\ln\frac{1}{\alpha}}$, $m$ is the number of the samples ($m = 100$ in our setting), and $\alpha$ is the one-side confidence parameter ($\alpha = 0.05$ in our setting). The proof of this estimation can be found in Appendix A.11.

### A.9.3 ESTIMATING THE REWARD LOWER BOUND FOR SMOOTHED AGENTS

In practice, we use Monte Carlo sampling to estimate $\widetilde{F}_{\pi,p}$, which denotes as $\widetilde{F}_{\pi,p_{\text{est}}}$. The estimation of the Reward Lower Bound is formulated as follows:

$$\widetilde{F}_{\pi,p_{\text{est}}}(\boldsymbol{\Delta s}) \geq \widetilde{F}_{\pi,\underline{p_{\text{est}}}}(\boldsymbol{0}), \ \text{s.t.} \ ||\boldsymbol{\Delta s}||_2 \leq B, \tag{16}$$

where $\widetilde{F}_{\pi,p_{\text{est}}}(\boldsymbol{\Delta s}) = max\{r \in \mathbb{R}||\{x \in S|x \leq r\}| \leq \lceil m_\tau p_{\text{est}}\rceil\}, S = \{F_\pi(\boldsymbol{\delta}_1 + \boldsymbol{\Delta s}), ..., F_\pi(\boldsymbol{\delta}_{m_\tau} + \boldsymbol{\Delta s})\}, \boldsymbol{\delta}_i \sim \mathcal{N}(0, \sigma^2 I_{H \times N}), \forall i \in \{1, ..., m_\tau\}, \underline{p_{\text{est}}} = \Phi(\Phi^{-1}(p_{\text{est}} - \Delta) - \frac{B}{\sigma}), \Delta = \sqrt{\frac{1}{2m_\tau}\ln\frac{1}{\alpha}}, m_\tau$ is the number of sample trajectories ($m_\tau = 1000$ in our setting), and $\alpha$ is the one-side confidence parameter ($\alpha = 0.05$ in our setting). Note that in this setting, each state is added with a perturbation. Therefore, $m = 1$. The proof of this estimation can be found in Appendix A.12.

## A.10 Proof of the certified radius for DS-DQN

In this section, we give the formal proof of the certified radius introduced in Section 4. Our proof is based on the proof proposed by Salman et al. (2019) in Appendix A. Recall that we have:

$$R_t = \frac{\sigma}{2}(\Phi^{-1}(\widetilde{Q}(s_t, a_1)) - \Phi^{-1}(\widetilde{Q}(s_t, a_2))), \tag{17}$$

where $a_1$ is the action with the largest Q-value among all the other actions, $a_2$ is the "runner-up" action, $R_t$ is the certified radius at time $t$, $\Phi$ is the CDF of normal distribution, $\sigma$ is the smoothing variance, and $\widetilde{Q}(s, a)$ is defined in Eq.(7).

We first go over the lemma needed for proof.

**Lemma 1** For the function $Q_h : \mathcal{S} \times \mathcal{A} \to [0, 1]$, the function $\widetilde{Q}$ is $\frac{1}{\sigma}\sqrt{\frac{2}{\pi}}$-Lipschitz.

*Proof.* From the definition of $\widetilde{Q}$, we have

$$\widetilde{Q}(s, a) = (Q_h * \mathcal{N}(0, \sigma^2 I_n))(D(s), a) = \frac{1}{(2\pi)^{n/2}\sigma^n} \int_{\mathbb{R}_n} Q_h(D(t), a) \exp\left(-\frac{1}{2\sigma^2}||s - t||_2^2\right) dt. \tag{18}$$

Take the gradient w.r.t. s, we have

$$\nabla_s \widetilde{Q}(s, a) = \frac{1}{(2\pi)^{n/2}\sigma^n} \int_{\mathbb{R}_n} \frac{1}{\sigma^2}(s - t)Q_h(D(t), a) \exp\left(-\frac{1}{2\sigma^2}||s - t||_2^2\right) dt. \tag{19}$$

For any unit direction $u$, we have

$$\begin{aligned}
u \cdot \nabla_s \widetilde{Q}(s, a) &\le \frac{1}{(2\pi)^{n/2}\sigma^n} \int_{\mathbb{R}_n} \frac{1}{\sigma^2}|u \cdot (s - t)| \exp\left(-\frac{1}{2\sigma^2}||s - t||_2^2\right) dt \\
&= \frac{1}{\sigma^2} \int_{\mathbb{R}_n} \frac{1}{\sqrt{2\pi}\sigma}|u \cdot (s - t)| \exp\left(-\frac{1}{2\sigma^2}||s - t||_2^2\right) dt \\
&= \frac{1}{\sigma^2} \int_{-\infty}^{+\infty} \frac{1}{\sqrt{2\pi}\sigma}|z| \exp\left(-\frac{1}{2\sigma^2}z^2\right) dz \\
&= \frac{1}{\sigma^2}\mathbb{E}_{z \sim \mathcal{N}(0,\sigma^2)}[|z|] \\
&= \frac{1}{\sigma}\sqrt{\frac{2}{\pi}}.
\end{aligned} \tag{20}$$

In fact, there is a stronger smoothness property for $\widetilde{Q}$.

**Lemma 2** For the function $Q_h : \mathcal{S} \times \mathcal{A} \to [0, 1]$, the mapping $s \mapsto \sigma\Phi^{-1}(\widetilde{Q}(s, a))$ is 1-Lipschitz.

*Proof.* Take the gradient of $\Phi^{-1}(\widetilde{Q}(s, a))$ w.r.t. s, we have

$$\nabla\Phi^{-1}(\widetilde{Q}(s, a)) = \frac{\nabla\widetilde{Q}(s, a)}{\Phi'(\Phi^{-1}(\widetilde{Q}(s, a)))}. \tag{21}$$

We intend to show that for any unit direction $u$,

$$\begin{aligned}
u \cdot \sigma\nabla\Phi^{-1}(\widetilde{Q}(s, a)) &\le 1 \\
u \cdot \sigma\nabla\widetilde{Q}(s, a) &\le \Phi'(\Phi^{-1}(\widetilde{Q}(s, a))) \\
u \cdot \sigma\nabla\widetilde{Q}(s, a) &\le \frac{1}{\sqrt{2\pi}} \exp\left(-\frac{1}{2}(\Phi^{-1}(\widetilde{Q}(s, a)))^2\right).
\end{aligned} \tag{22}$$

The left-hand side can be written as

$$\frac{1}{\sigma}\mathbb{E}_{\delta \sim \mathcal{N}(0,\sigma^2 I_n)}[Q_h(D(s + \delta), a)\delta \cdot u]. \tag{23}$$

We claim that the supremum of the above quantity over all functions $Q_h : \mathcal{S} \times \mathcal{A} \to [0, 1]$, subject to $\mathbb{E}[Q_h(D(s + \delta), a)] = \widetilde{Q}(s, a)$, is equal to

$$\frac{1}{\sigma}\mathbb{E}[(\delta \cdot u)\mathbb{1}\{\delta \cdot u \geq -\sigma\Phi^{-1}(\widetilde{Q}(s,a))\}] = \frac{1}{\sqrt{2\pi}}\exp\left(-\frac{1}{2}(\Phi^{-1}(\widetilde{Q}(s,a)))^2\right). \tag{24}$$

To prove the claim is true, note that $h : \delta \mapsto \mathbb{1}\{\delta \cdot u \geq -\sigma\Phi^{-1}(\widetilde{Q}(s,a))\}$ achieves equality. Assume by contradiction that the maximum is reached by some function $f : \delta \to [0, 1]$. Consider the set $\Omega^+ = \{\delta | h(\delta) > f(\delta)\}$ and the set $\Omega^- = \{\delta | h(\delta) < f(\delta)\}$. Now construct the new function $f' = f + (h - f)\mathbb{1}\{\Omega^+\} - (f - h)\mathbb{1}\{\Omega^-\}$, which takes value in $[0, 1]$. Since both $h$ and $f$ integrate to $\widetilde{Q}(s, a)$, we have $\int_{\Omega^+}(h - f)d\delta = \int_{\Omega^-}(f - h)d\delta$. This gives that $f'$ also integrates to $\widetilde{Q}(s, a)$. By the definition of $h$, for any $\delta_1 \in \Omega^+$ and $\delta_2 \in \Omega^-$, we have $\delta_1 \cdot u > \delta_2 \cdot u$, and since $\int_{\Omega^+}(h - f)d\delta = \int_{\Omega^-}(f - h)d\delta$, we have

$$\int_{\Omega^+}(\delta \cdot u)(h - f)(\delta)d\delta > \int_{\Omega^-}(\delta \cdot u)(f - h)(\delta)d\delta$$

$$\int(\delta \cdot u)f(\delta)d\delta < \int(\delta \cdot u)f(\delta)d\delta + \int_{\Omega^+}(\delta \cdot u)(h - f)(\delta)d\delta - \int_{\Omega^-}(\delta \cdot u)(f - h)(\delta)d\delta \tag{25}$$

$$\int(\delta \cdot u)f(\delta)d\delta < \int(\delta \cdot u)f'(\delta)d\delta$$

Hence, the maximum is obtained at $h$. The claim holds, and hence, we have

$$u \cdot \sigma\nabla\Phi^{-1}(\widetilde{Q}(s,a)) \leq 1. \tag{26}$$

Now, we can prove the certified radius in Eq.(17).

**Theorem 1** Let $Q_h : \mathcal{S} \times \mathcal{A} \to [0, 1]$, and $\widetilde{Q}(s, a) = \mathbb{E}_{\delta \sim \mathcal{N}(0,\sigma^2 I)}Q_h(D(s + \delta), a)$. At time step $t$ with state $s_t$, the certified radius is

$$R_t = \frac{\sigma}{2}(\Phi^{-1}(\widetilde{Q}(s_t, a_1)) - \Phi^{-1}(\widetilde{Q}(s_t, a_2))), \tag{27}$$

where $a_1$ is the action with the largest Q-value among all the other actions, $a_2$ is the "runner-up" action, $R_t$ is the certified radius at time $t$, $\Phi$ is the CDF of normal distribution, and $\sigma$ is the smoothing variance. The certified radius gives a lower bound on the minimum $\ell_2$ adversarial perturbation required to change the policy from $a_1$ to $a_2$.

*Proof.* Let the perturbation be $\Delta s$ and able to change the action from $a_1$ to $a_2$. By lemma 2, we have

$$\sigma\Phi^{-1}(\widetilde{Q}(s_t, a_1)) - \sigma\Phi^{-1}(\widetilde{Q}(s_t + \Delta s, a_1)) \leq ||\Delta s||_2 \tag{28}$$

Since the perturbation can change the action, we have $\widetilde{Q}(s_t + \Delta s, a_1) \leq \widetilde{Q}(s_t + \Delta s, a_2)$, which leads to

$$\sigma\Phi^{-1}(\widetilde{Q}(s_t, a_1)) - \sigma\Phi^{-1}(\widetilde{Q}(s_t + \Delta s, a_2)) \leq ||\Delta s||_2 \tag{29}$$

By lemma 2 and $\widetilde{Q}(s_t + \Delta s, a_2) \geq \widetilde{Q}(s_t, a_2)$, we have

$$\sigma\Phi^{-1}(\widetilde{Q}(s_t + \Delta s, a_2)) - \sigma\Phi^{-1}(\widetilde{Q}(s_t, a_2)) \leq ||\Delta s||_2 \tag{30}$$

Combine Eq.(29) and Eq.(30), we have

$$||\Delta s||_2 \geq \frac{\sigma}{2}(\Phi^{-1}(\widetilde{Q}(s_t, a_1)) - \Phi^{-1}(\widetilde{Q}(s_t, a_2))), \tag{31}$$

which gives us the certified radius

$$R_t = \frac{\sigma}{2}(\Phi^{-1}(\widetilde{Q}(s_t, a_1)) - \Phi^{-1}(\widetilde{Q}(s_t, a_2))). \tag{32}$$

Now, we prove the practical version of the certified radius introduced in Appendix A.9.1:

**Theorem 2** Let $Q_h : \mathcal{S} \times \mathcal{A} \to [0,1]$, and $\widetilde{Q}_{\text{est}}(s,a) = \frac{1}{m}\Sigma_{i=1}^m Q_h(D(s+\delta_i),a), \delta_i \sim \mathcal{N}(0, \sigma^2 I_N), \forall i \in \{1,...,m\}$. At time step $t$ with state $s_t$, the certified radius is

$$R_{\text{est},t} = \frac{\sigma}{2}(\Phi^{-1}(\widetilde{Q}_{\text{est}}(s_t,a_1)-\Delta) - \Phi^{-1}(\widetilde{Q}_{\text{est}}(s_t,a_2)+\Delta)), \tag{33}$$

where $\Delta = \sqrt{\frac{1}{2m}\ln\frac{1}{\alpha}}$, $m$ is the number of the samples, $\alpha$ is the one-side confidence parameter, $a_1$ is the action with the largest Q-value among all the other actions, $a_2$ is the "runner-up" action, $R_t$ is the certified radius at time $t$, $\Phi$ is the CDF of normal distribution, and $\sigma$ is the smoothing variance.

*Proof.* By *Hoeffding's Inequality*, for any $t \geq 0$, we have

$$P(\widetilde{Q}_{\text{est}} - \widetilde{Q} \geq t) \leq \exp^{-2mt^2}. \tag{34}$$

Rearrange the inequality

$$P(\widetilde{Q}_{\text{est}} - \widetilde{Q} \geq \sqrt{\frac{1}{2m}\ln\frac{1}{\alpha}}) \leq \alpha. \tag{35}$$

Hence, a $1-\alpha$ confidence lower bound $\underline{\widetilde{Q}}$ of $\widetilde{Q}$ is

$$\underline{\widetilde{Q}} = \widetilde{Q}_{\text{est}} - \sqrt{\frac{1}{2m}\ln\frac{1}{\alpha}} = \widetilde{Q}_{\text{est}} - \Delta. \tag{36}$$

Similarly, we have $1-\alpha$ confidence upper bound $\overline{\widetilde{Q}}$ of $\widetilde{Q}$

$$\overline{\widetilde{Q}} = \widetilde{Q}_{\text{est}} + \Delta. \tag{37}$$

Substitute $\widetilde{Q}(s_t,a_1)$ with the lower bound and $\widetilde{Q}(s_t,a_2)$ with the upper bound, we have

$$R_{\text{est},t} = \frac{\sigma}{2}(\Phi^{-1}(\widetilde{Q}_{\text{est}}(s_t,a_1)-\Delta) - \Phi^{-1}(\widetilde{Q}_{\text{est}}(s_t,a_2)+\Delta)) \tag{38}$$

### A.11 Proof of the action bound for AS-PPO

In this section, we give the formal proof of the action bound introduced in Section 4. Our proof is based on the proof proposed by Chiang et al. (2020) in Appendix B. Recall that we have:

$$\tilde{\pi}_{\mathrm{det},\underline{p}}(s_t) \preceq \tilde{\pi}_{\mathrm{det},p}(s_t + \Delta s) \preceq \tilde{\pi}_{\mathrm{det},\overline{p}}(s_t), \ s.t \ ||\Delta s||_2 \leq \epsilon, \tag{39}$$

where $\tilde{\pi}_{i,\mathrm{det},p}(s) = \sup\{a_i \in \mathbb{R}|\mathbb{P}_{\delta\sim\mathcal{N}(0,\sigma^2 I)}[\pi_{i,\mathrm{det}}(s+\delta) \leq a_i] \leq p\}, \forall i \in \{1,...,N_{\mathrm{action}}\}$, $\underline{p} = \Phi(\Phi^{-1}(p) - \frac{\epsilon}{\sigma}), \overline{p} = \Phi(\Phi^{-1}(p) + \frac{\epsilon}{\sigma})$, $\Phi$ is the CDF of normal distribution, and $\sigma$ is the smoothing variance.

**Theorem 3** Let $\pi : \mathcal{S} \to \mathcal{A}$ be the policy network, and $\tilde{\pi}_{i,\mathrm{det},p}(s) = \sup\{a_i \in \mathbb{R}|\mathbb{P}_{\delta\sim\mathcal{N}(0,\sigma^2 I)}[\pi_{i,\mathrm{det}}(s+\delta) \leq a_i] \leq p\}, \forall i \in \{1,...,N_{\mathrm{action}}\}$. At time step $t$ with state $s_t$, the action bound is

$$\tilde{\pi}_{\mathrm{det},\underline{p}}(s_t) \preceq \tilde{\pi}_{\mathrm{det},p}(s_t + \Delta s) \preceq \tilde{\pi}_{\mathrm{det},\overline{p}}(s_t), \ s.t \ ||\Delta s||_2 \leq \epsilon, \tag{40}$$

where $\underline{p} = \Phi(\Phi^{-1}(p) - \frac{\epsilon}{\sigma}), \overline{p} = \Phi(\Phi^{-1}(p) + \frac{\epsilon}{\sigma})$, $\Phi$ is the CDF of a normal distribution, and $\sigma$ is the smoothing variance.

*Proof.* Let $\mathcal{E}_i(s_t) = \mathbb{E}_{\delta\sim\mathcal{N}(0,\sigma^2 I_N)}[\mathbb{1}\{\pi_{i,\mathrm{det}}(s_t+\delta) \leq \tilde{\pi}_{i,\mathrm{det},\underline{p}}(s_t)\}]$, and we have $\mathcal{E}_i : \mathbb{R}^N \to [0,1]$, $\forall i \in \{1,...,N_{\mathrm{action}}\}$. The mapping $s_t \mapsto \sigma\Phi^{-1}(\mathcal{E}_i(s_t))$ is 1-Lipschitz, which can be proved by the similar technique used in Lemma 2. Since $\mathcal{E}_i(s_t) = \mathbb{P}_{\delta\sim\mathcal{N}(0,\sigma^2 I_N)}[\pi_{i,\mathrm{det}}(s_t+\delta) \leq \tilde{\pi}_{i,\mathrm{det},\underline{p}}(s_t)]$, given the perturbation $\Delta s$, we have

$$\sigma\Phi^{-1}(\mathbb{P}_{\delta\sim\mathcal{N}(0,\sigma^2 I_N)}[\pi_{i,\mathrm{det}}(s_t+\delta+\Delta s) \leq \tilde{\pi}_{i,\mathrm{det},\underline{p}}(s_t)])-$$
$$\sigma\Phi^{-1}(\mathbb{P}_{\delta\sim\mathcal{N}(0,\sigma^2 I_N)}[\pi_{i,\mathrm{det}}(s_t+\delta) \leq \tilde{\pi}_{i,\mathrm{det},\underline{p}}(s_t)]) \leq ||\Delta s||_2. \tag{41}$$

Rearrange the inequality, we have

$$\Phi^{-1}(\mathbb{P}_{\delta\sim\mathcal{N}(0,\sigma^2 I_N)}[\pi_{i,\mathrm{det}}(s_t+\delta+\Delta s) \leq \tilde{\pi}_{i,\mathrm{det},\underline{p}}(s_t)])$$
$$\leq \Phi^{-1}(\mathbb{P}_{\delta\sim\mathcal{N}(0,\sigma^2 I_N)}[\pi_{i,\mathrm{det}}(s_t+\delta) \leq \tilde{\pi}_{i,\mathrm{det},\underline{p}}(s_t)]) + \frac{||\Delta s||_2}{\sigma}$$
$$\leq \Phi^{-1}(\mathbb{P}_{\delta\sim\mathcal{N}(0,\sigma^2 I_N)}[\pi_{i,\mathrm{det}}(s_t+\delta) \leq \tilde{\pi}_{i,\mathrm{det},\underline{p}}(s_t)]) + \frac{\epsilon}{\sigma} \tag{42}$$
$$= \Phi^{-1}(\underline{p}) + \frac{\epsilon}{\sigma}$$
$$= \Phi^{-1}(p).$$

By the monotonicity of $\Phi$, we have

$$\mathbb{P}_{\delta\sim\mathcal{N}(0,\sigma^2 I_N)}[\pi_{i,\mathrm{det}}(s_t+\delta+\Delta s) \leq \tilde{\pi}_{i,\mathrm{det},\underline{p}}(s_t)] \leq p. \tag{43}$$

Recall that $\tilde{\pi}_{i,\mathrm{det},p}(s_t + \Delta s) = \sup\{a_i \in \mathbb{R}|\mathbb{P}_{\delta\sim\mathcal{N}(0,\sigma^2 I_N)}[\pi_{i,\mathrm{det}}(s_t+\delta+\Delta s) \leq a_i] \leq p\}, \forall i \in \{1,...,N_{\mathrm{action}}\}$, we have

$$\tilde{\pi}_{\mathrm{det},\underline{p}}(s_t) \preceq \tilde{\pi}_{\mathrm{det},p}(s_t + \Delta s). \tag{44}$$

We can show that $\tilde{\pi}_{\mathrm{det},p}(s_t + \Delta s) \preceq \tilde{\pi}_{\mathrm{det},\overline{p}}(s_t)$ for all $||\Delta s||_2 \leq \epsilon$ with the similar technique. Combine the two bounds we have

$$\tilde{\pi}_{\mathrm{det},\underline{p}}(s_t) \preceq \tilde{\pi}_{\mathrm{det},p}(s_t + \Delta s) \preceq \tilde{\pi}_{\mathrm{det},\overline{p}}(s_t). \tag{45}$$

Now, we prove the practical version of the action bound introduced in Appendix A.9.2:

**Theorem 4** Let $\pi : \mathcal{S} \to \mathcal{A}$ be the policy network, and $\tilde{\pi}_{i,\mathrm{det},p_{\mathrm{est}}}(s) = max\{a_i \in \mathbb{R}| \ |\{x \in S_i|x \leq a_i\}| \leq \lceil mp_{\mathrm{est}}\rceil\}, S_i = \{\pi_{i,\mathrm{det}}(s+\delta_1),...,\pi_{i,\mathrm{det}}(s+\delta_m)\}, \forall i \in \{1,...,N_{\mathrm{action}}\}, \delta_j \sim \mathcal{N}(0,\sigma^2 I_N), \forall j = 1,...,m$. At time step $t$ with state $s_t$, the action bound is

$$\tilde{\pi}_{\mathrm{det},\underline{p_{\mathrm{est}}}}(s_t) \preceq \tilde{\pi}_{\mathrm{det},p_{\mathrm{est}}}(s_t + \Delta s) \preceq \tilde{\pi}_{\mathrm{det},\overline{p_{\mathrm{est}}}}(s_t), \ s.t \ ||\Delta s||_2 \leq \epsilon, \tag{46}$$

where $\underline{p_{\mathrm{est}}} = \Phi(\Phi^{-1}(p_{\mathrm{est}} - \Delta) - \frac{\epsilon}{\sigma}), \overline{p_{\mathrm{est}}} = \Phi(\Phi^{-1}(p_{\mathrm{est}} + \Delta) + \frac{\epsilon}{\sigma}), \Delta = \sqrt{\frac{1}{2m}\ln\frac{1}{\alpha}}$, $m$ is the number of the samples, $\alpha$ is the one-side confidence parameter, $\Phi$ is the CDF of normal distribution, and $\sigma$ is the smoothing variance.

*Proof.* By *Hoeffding's Inequality*, for any $t \geq 0$, we have

$$P(p_{\text{est}} - p \geq t) \leq \exp^{-2mt^2}. \tag{47}$$

Rearrange the inequality

$$P(p_{\text{est}} - p \geq \sqrt{\frac{1}{2m} \ln \frac{1}{\alpha}}) \leq \alpha. \tag{48}$$

Hence, a $1 - \alpha$ confidence lower bound $\underline{p}$ of $p$ is

$$\underline{p} = p_{\text{est}} - \sqrt{\frac{1}{2m} \ln \frac{1}{\alpha}} = p_{\text{est}} - \Delta. \tag{49}$$

Similarly, we have $1 - \alpha$ confidence upper bound $\overline{p}$ of $\underline{p}$

$$\overline{p} = p_{\text{est}} + \Delta. \tag{50}$$

Substitute $\Phi(\Phi^{-1}(p) - \frac{\epsilon}{\sigma})$ with the lower bound, and $\Phi(\Phi^{-1}(p) + \frac{\epsilon}{\sigma})$ with the upper bound, we have

$$\left[ \Phi(\Phi^{-1}(p_{\text{est}} - \Delta) - \frac{\epsilon}{\sigma}), \quad \Phi(\Phi^{-1}(p_{\text{est}} + \Delta) + \frac{\epsilon}{\sigma}) \right], \tag{51}$$

which is the new upper bound and lower bound in the expression.

A.12 PROOF OF THE REWARD LOWER BOUND FOR SMOOTHED AGENTS

In this section, we give the formal proof of the reward lower bound introduced in Section 4. Our proof is based on the proof proposed by Chiang et al. (2020) in Appendix B. Recall that we have:

$$\widetilde{F}_{\pi,p}(\boldsymbol{\Delta s}) \geq \widetilde{F}_{\pi,\underline{p}}(\mathbf{0}), \text{ s.t. } ||\boldsymbol{\Delta s}||_2 \leq B, \tag{52}$$

where $\widetilde{F}_{\pi,p}(\boldsymbol{\Delta s}) = \sup\{r \in \mathbb{R}|\mathbb{P}_{\boldsymbol{\delta}\sim\mathcal{N}(0,\sigma^2 I_{H\times N})}[F_\pi(\boldsymbol{\delta} + \boldsymbol{\Delta s}) \leq r] \leq p\}$, $\underline{p} = \Phi(\Phi^{-1}(p) - \frac{B}{\sigma})$, and $B$ is the $\ell_2$ attack budget of the entire trajectory.

**Theorem 5** Let $F_\pi : \mathbb{R}^{H\times N} \to \mathbb{R}$ be the function mapping the perturbation to the total reward, and $\widetilde{F}_{\pi,p}(\boldsymbol{\Delta s}) = \sup\{r \in \mathbb{R}|\mathbb{P}_{\boldsymbol{\delta}\sim\mathcal{N}(0,\sigma^2 I_{H\times N})}[F_\pi(\boldsymbol{\delta} + \boldsymbol{\Delta s}) \leq r] \leq p\}$. The reward lower bound is

$$\widetilde{F}_{\pi,p}(\boldsymbol{\Delta s}) \geq \widetilde{F}_{\pi,\underline{p}}(\mathbf{0}), \text{ s.t. } ||\boldsymbol{\Delta s}||_2 \leq B, \tag{53}$$

where $\underline{p} = \Phi(\Phi^{-1}(p) - \frac{B}{\sigma})$, $B$ is the $\ell_2$ attack budget of the entire trajectory, $\Phi$ is the CDF of normal distribution, and $\sigma$ is the smoothing variance.

*Proof.* Let $\mathcal{E}(\boldsymbol{\Delta s}) = \mathbb{E}_{\boldsymbol{\delta}\sim\mathcal{N}(0,\sigma^2 I_{H\times N})}[\mathbb{1}\{F_\pi(\boldsymbol{\delta} + \boldsymbol{\Delta s}) \leq \widetilde{F}_{\pi,\underline{p}}(\mathbf{0})\}]$, and we have $\mathcal{E} : \mathbb{R}^{H\times N} \to [0,1]$. The mapping $\boldsymbol{\Delta s} \mapsto \sigma\Phi^{-1}(\mathcal{E}(\boldsymbol{\Delta s}))$ is 1-Lipschitz by Lemma 2. Since $\mathcal{E}(\boldsymbol{\Delta s}) = \mathbb{P}_{\boldsymbol{\delta}\sim\mathcal{N}(0,\sigma^2 I_{H\times N})}[F_\pi(\boldsymbol{\delta} + \boldsymbol{\Delta s}) \leq \widetilde{F}_{\pi,\underline{p}}(\mathbf{0})]$, given the perturbation $\boldsymbol{\Delta s}$, we have

$$\sigma\Phi^{-1}(\mathbb{P}_{\boldsymbol{\delta}\sim\mathcal{N}(0,\sigma^2 I_{H\times N})}[F_\pi(\boldsymbol{\delta} + \boldsymbol{\Delta s}) \leq \widetilde{F}_{\pi,\underline{p}}(\mathbf{0})]) - \sigma\Phi^{-1}(\mathbb{P}_{\boldsymbol{\delta}\sim\mathcal{N}(0,\sigma^2 I_{H\times N})}[F_\pi(\boldsymbol{\delta}) \leq \widetilde{F}_{\pi,\underline{p}}(\mathbf{0})])$$
$$\leq ||\boldsymbol{\Delta s}||_2. \tag{54}$$

Rearrange the inequality, we have

$$\Phi^{-1}(\mathbb{P}_{\boldsymbol{\delta}\sim\mathcal{N}(0,\sigma^2 I_{H\times N})}[F_\pi(\boldsymbol{\delta} + \boldsymbol{\Delta s}) \leq \widetilde{F}_{\pi,\underline{p}}(\mathbf{0})])$$
$$\leq \Phi^{-1}(\mathbb{P}_{\boldsymbol{\delta}\sim\mathcal{N}(0,\sigma^2 I_{H\times N})}[F_\pi(\boldsymbol{\delta}) \leq \widetilde{F}_{\pi,\underline{p}}(\mathbf{0})]) + \frac{||\boldsymbol{\Delta s}||_2}{\sigma}$$
$$\leq \Phi^{-1}(\mathbb{P}_{\boldsymbol{\delta}\sim\mathcal{N}(0,\sigma^2 I_{H\times N})}[F_\pi(\boldsymbol{\delta}) \leq \widetilde{F}_{\pi,\underline{p}}(\mathbf{0})]) + \frac{B}{\sigma} \tag{55}$$
$$= \Phi^{-1}(\underline{p}) + \frac{B}{\sigma}$$
$$= \Phi^{-1}(p).$$

By the monotonicity of $\Phi$, we have

$$\mathbb{P}_{\boldsymbol{\delta}\sim\mathcal{N}(0,\sigma^2 I_{H\times N})}[F_\pi(\boldsymbol{\delta} + \boldsymbol{\Delta s}) \leq \widetilde{F}_{\pi,\underline{p}}(\mathbf{0})] \leq p. \tag{56}$$

Recall that $\widetilde{F}_{\pi,p}(\boldsymbol{\Delta s}) = \sup\{r \in \mathbb{R}|\mathbb{P}_{\boldsymbol{\delta}\sim\mathcal{N}(0,\sigma^2 I_{H\times N})}[F_\pi(\boldsymbol{\delta} + \boldsymbol{\Delta s}) \leq r] \leq p\}$, we have

$$\widetilde{F}_{\pi,p}(\boldsymbol{\Delta s}) \geq \widetilde{F}_{\pi,\underline{p}}(\mathbf{0}). \tag{57}$$

Now, we prove the practical version of the reward lower bound introduced in Appendix A.9.3:

**Theorem 6** Let $F_\pi : \mathbb{R}^{H\times N} \to \mathbb{R}$ be the function mapping the perturbation to the total reward, and $\widetilde{F}_{\pi,p_{\text{est}}}(\boldsymbol{\Delta s}) = max\{r \in \mathbb{R}||\{x \in S|x \leq r\}| \leq \lceil m_\tau p_{\text{est}}\rceil\}, S = \{F_\pi(\boldsymbol{\delta}_1 + \boldsymbol{\Delta s}), ..., F_\pi(\boldsymbol{\delta}_{m_\tau} + \boldsymbol{\Delta s})\}, \boldsymbol{\delta}_i \sim \mathcal{N}(0, \sigma^2 I_{H\times N}), \forall i = \{1, ..., m_\tau\}$. The reward lower bound is

$$\widetilde{F}_{\pi,p_{\text{est}}}(\boldsymbol{\Delta s}) \geq \widetilde{F}_{\pi,\underline{p_{\text{est}}}}(\mathbf{0}), \text{ s.t. } ||\boldsymbol{\Delta s}||_2 \leq B, \tag{58}$$

where $\underline{p_{\text{est}}} = \Phi(\Phi^{-1}(p_{\text{est}} - \Delta) - \frac{B}{\sigma}), \Delta = \sqrt{\frac{1}{2m_\tau}\ln\frac{1}{\alpha}}$, $m_\tau$ is the number of sample trajectories, $\alpha$ is the one-side confidence parameter, $\Phi$ is the CDF of normal distribution, and $\sigma$ is the smoothing variance.

*Proof.* By *Hoeffding's Inequality*, for any $t \geq 0$, we have

$$P(p_{\text{est}} - p \geq t) \leq \exp^{-2m_\tau t^2}. \tag{59}$$

Rearrange the inequality

$$P(p_{\text{est}} - p \geq \sqrt{\frac{1}{2m_\tau} \ln \frac{1}{\alpha}}) \leq \alpha. \tag{60}$$

Hence, a $1 - \alpha$ confidence lower bound $\underline{p}$ of $p$ is

$$\underline{p} = p_{\text{est}} - \sqrt{\frac{1}{2m_\tau} \ln \frac{1}{\alpha}} = p_{\text{est}} - \Delta. \tag{61}$$

Substitute $\Phi(\Phi^{-1}(p) - \frac{B}{\sigma})$ with the lower bound, we have

$$\Phi(\Phi^{-1}(p_{\text{est}} - \Delta) - \frac{B}{\sigma}), \tag{62}$$

which is the new lower bound in the expression.

### A.13 THE ACTION DIVERGENCE OF SMOOTHED PPO AGENTS

We designed a metric based on the action bound in Section 4 to evaluate the certified robustness of the smoothed PPO agents. We define the **Action Divergence** as follows:

$$\text{ADIV} = \mathbb{E}_{s,\epsilon}\left[\frac{||\tilde{\pi}_{\text{det},\overline{p_{\text{est}}}}(s) - \tilde{\pi}_{\text{det},\underline{p_{\text{est}}}}(s)||_2}{2\epsilon}\right], \tag{63}$$

where $\epsilon$ is the $\ell_2$ attack budget used in estimating the action bound, and the definition of $\overline{p_{\text{est}}}$ and $\underline{p_{\text{est}}}$ is in Appendix A.9.2. We found that the $\ell_2$ norm of the difference between the upper and lower bound of the actions is proportional to the magnitude of the $\ell_2$ budget $\epsilon$, which makes $\frac{||\tilde{\pi}_{\text{det},\overline{p_{\text{est}}}}(s) - \tilde{\pi}_{\text{det},\underline{p_{\text{est}}}}(s)||_2}{2\epsilon}$ almost unchanged under different $\epsilon$ setting. Hence, we take the expectation over the state $s$ and the budget $\epsilon$ to estimate this fraction, which is the ADIV proposed here. We estimate the ADIV by taking the average of 50 trajectories with three different $\epsilon$ settings ($\epsilon = 0.1$, $\epsilon = 0.2$, and $\epsilon = 0.3$).

ADIV describes the worst-case stability of the actions of a PPO smoothed agent under any $\ell_2$ perturbation. The more this value is, the more unstable the smoothed agent is under the $\ell_2$ attack. The result is shown in Table 6. Generally, all the smoothed robust agents have a smaller ADIV than the smoothed vanillaPPO agent. Note that although the SAPPO+RS (Convex) implementation has the smallest ADIV, our AS-PPO performs better under Min-RS attack and has a higher reward lower bound compared to SAPPO+RS (Convex) as shown in Table 3

Table 6: The Action Divergence of different smoothed agents.

| Methods | Action Divergence (lower is better) | | |
|---|---|---|---|
| | Walker | Hopper | Humanoid |
| AS-PPO (Ours) | 4.199 | 1.378 | 3.257 |
| SAPPO+RS (SGLD) | 2.836 | 1.773 | 3.095 |
| SAPPO+RS (Convex) | **1.258** | **1.183** | **1.852** |
| VanillaPPO+RS | 5.090 | 4.650 | 8.698 |

A.14 DETAIL EXPERIMENT RESULTS OF REWARD LOWER BOUND

Table 7 shows the details of the reward lower bound for smoothed DQN agents under different $\ell_2$ budget $\epsilon$. We use the same budget $\epsilon$ for every state, and hence, the total budget $B = \epsilon\sqrt{H}$, where $H$ is the length of the trajectory. We set $H = 2500$ in Pong, Freeway, and RoadRunner. The reward lower bound of DS-DQN (Vanilla) is comparable with the bound of DS-DQN (Radial), DS-DQN (SADQN), and DAS-DQN (Vanilla), which suggests that our Denoised Smoothing setting already achieved a high robustness guarantee even without further using other robust agents as base models or leveraging adversarial training.

Table 7: The reward lower bound of different smoothed DQN agents under different $\ell_2$ attack budgets. The smoothing variance $\sigma$ for all the agents is set to 0.1 in Pong, 0.12 in Freeway, and $\sigma = 0.1$ in RoadRunner.

| Pong | $\ell_2$ attack budget | | | | |
|---|---|---|---|---|---|
| $\epsilon(\ell_2)$ | 0.001 | 0.002 | 0.003 | 0.004 | 0.005 |
| **Ours:** | | | | | |
| DS-DQN (Vanilla) | 18.0 | 17.0 | 16.0 | 14.0 | 12.0 |
| DS-DQN (Radial) | 20.0 | 20.0 | 19.0 | 19.0 | 19.0 |
| DS-DQN (SADQN) | **21.0** | **21.0** | **21.0** | **20.0** | **19.7** |
| DAS-DQN (Vanilla) | 18.0 | 17.0 | 15.0 | 14.0 | 11.0 |
| **CROP:** | | | | | |
| CROP (Radial) | $-21.0$ | $-21.0$ | $-21.0$ | $-21.0$ | $-21.0$ |
| CROP (SADQN) | $-21.0$ | $-21.0$ | $-21.0$ | $-21.0$ | $-21.0$ |
| CROP (Vanilla) | $-21.0$ | $-21.0$ | $-21.0$ | $-21.0$ | $-21.0$ |
| **Naive training with RS:** | | | | | |
| S-DQN | $-21.0$ | $-21.0$ | $-21.0$ | $-21.0$ | $-21.0$ |
| Freeway | | | | | |
| **Ours:** | | | | | |
| DS-DQN (Vanilla) | 28.0 | 28.0 | 27.0 | 26.0 | **26.0** |
| DS-DQN (Radial) | 28.0 | 27.0 | 26.0 | 26.0 | 25.0 |
| DS-DQN (SADQN) | 27.0 | 26.0 | 25.0 | 24.0 | 24.0 |
| DAS-DQN (Vanilla) | **29.0** | **29.0** | **28.0** | **27.0** | **26.0** |
| **CROP:** | | | | | |
| CROP (Radial) | 20.4 | 20.0 | 20.0 | 20.0 | 19.0 |
| CROP (SADQN) | 21.0 | 20.0 | 20.0 | 20.0 | 19.0 |
| CROP (Vanilla) | 11.0 | 10.0 | 10.0 | 9.0 | 8.0 |
| **Naive training with RS:** | | | | | |
| S-DQN | 0.0 | 0.0 | 0.0 | 0.0 | 0.0 |
| RoadRunner | | | | | |
| **Ours:** | | | | | |
| DS-DQN (Vanilla) | **25900.0** | **23900.0** | **22000.0** | 19400.0 | 16500.0 |
| DS-DQN (Radial) | 24500.0 | 22515.0 | 20800.0 | 19297.2 | 12920.9 |
| DS-DQN (SADQN) | 25200.0 | 23400.0 | **22000.0** | **20594.4** | **19169.5** |
| DAS-DQN (Vanilla) | 25100.0 | 23000.0 | 21100.0 | 19600.0 | 17769.5 |
| **CROP:** | | | | | |
| CROP (Radial) | 6400.0 | 5300.0 | 3800.0 | 3400.0 | 3000.0 |
| CROP (SADQN) | 11900.0 | 10700.0 | 8800.0 | 6600.0 | 3300.0 |
| CROP (Vanilla) | 0.0 | 0.0 | 0.0 | 0.0 | 0.0 |
| **Naive training with RS:** | | | | | |
| S-DQN | 900.0 | 900.0 | 900.0 | 900.0 | 900.0 |

Table 8 shows the reward lower bound for smoothed PPO agents under different $\ell_2$ budget $\epsilon$. We use the same budget $\epsilon$ for every state, and hence, the total budget $B = \epsilon\sqrt{H}$, where $H$ is the length of the trajectory. We set $H = 1000$ in Walker, Hopper, and Humanoid. Our AS-PPO exhibits a higher reward lower bound under Walker and Humanoid environments. Despite not outperforming SAPPO+RS (SGLD) in Hopper, AS-PPO still has a much higher reward lower bound compared to VanillaPPO+RS.

Table 8: The reward lower bound of different smoothed PPO agents under different $\ell_2$ attack budgets. The smoothing variance $\sigma$ for all the agents is set to 0.2 in Walker, 0.3 in Hopper, and $\sigma = 0.4$ in Humanoid.

| Walker | $\ell_2$ attack budget | | | | |
|---|---|---|---|---|---|
| $\epsilon(\ell_2)$ | 0.002 | 0.004 | 0.006 | 0.008 | 0.01 |
| **Ours:** | | | | | |
| AS-PPO | **5345.3** | **5013.5** | **4869.6** | 3255.9 | 2391.94 |
| **Naively smoothed agents:** | | | | | |
| SAPPO+RS (SGLD) | 4641.8 | 4545.2 | 4246.9 | **3382.6** | **2534.5** |
| SAPPO+RS (Convex) | 4307.2 | 4247.4 | 4149.5 | 3207.0 | 2367.2 |
| VanillaPPO+RS | 1474.7 | 1250.3 | 1118.1 | 894.1 | 630.7 |
| **Naive training with RS:** | | | | | |
| S-PPO | 4047.4 | 4001.7 | 3732.9 | 2244.8 | 1760.4 |
| Hopper | | | | | |
| **Ours:** | | | | | |
| AS-PPO | 2055.3 | 1828.4 | 1694.4 | 1526.7 | 1438.6 |
| **Naively smoothed agents:** | | | | | |
| SAPPO+RS (SGLD) | **2075.9** | **1891.2** | **1814.0** | **1693.6** | **1590.8** |
| SAPPO+RS (Convex) | 2012.1 | 1839.1 | 1768.1 | 1657.1 | 1485.9 |
| VanillaPPO+RS | 1084.5 | 1014.5 | 899.3 | 832.3 | 686.3 |
| **Naive training with RS:** | | | | | |
| S-PPO | 1731.19 | 1562.0 | 1439.5 | 1358.1 | 1254.5 |
| Humanoid | | | | | |
| **Ours:** | | | | | |
| AS-PPO | **7075.3** | **7065.0** | **7019.2** | **7008.9** | **7000.2** |
| **Naively smoothed agents:** | | | | | |
| SAPPO+RS (SGLD) | 6836.9 | 6829.8 | 6823.4 | 6817.0 | 6812.3 |
| SAPPO+RS (Convex) | 6435.2 | 6424.6 | 6415.5 | 6407.2 | 6398.8 |
| VanillaPPO+RS | 2815.3 | 2250.3 | 1984.9 | 1761.7 | 1572.5 |
| **Naive training with RS:** | | | | | |
| S-PPO | 6212.3 | 6207.0 | 6200.4 | 6195.3 | 6189.8 |

A.15   DETAIL EXPERIMENT RESULTS OF REWARD UNDER ATTACK

Table 9 and Table 10 shows the reward of DQN agents under $\ell_\infty$ and $\ell_2$ PGD attack. Note that we used our new attack, which is stronger than the classic PGD attack, in Section 3.1 to evaluate all the smoothed agents. Our DS-DQN (Vanilla) already outperformed the state-of-the-art robust agent, RadialDQN, in most of the settings except for $\ell_\infty$ attack in RoadRunner. The problem of not performing well under $\ell_\infty$ attack in RoadRunner was solved by introducing DS-DQN (Radial), DS-DQN (SADQN), and DAS-DQN. DS-DQN (Radial) performs especially well under PGD attack in all the environments, which suggests that our DS-DQN can be further boosted by changing the base model to a robust agent.

Table 9: The reward of DQN agents under $\ell_\infty$ PGD attack. The smoothing variance $\sigma$ for our agents is set to 0.1 in Pong, 0.12 in Freeway, and $\sigma = 0.1$ in RoadRunner. For all the CROP small $\sigma$ agents, $\sigma = 0.01$.

| Pong | $\ell_\infty$ PGD attack (used our new attack to evaluate the smoothed agents) | | | | |
|---|---|---|---|---|---|
| $\epsilon(\ell_\infty)$ | 0.01 | 0.02 | 0.03 | 0.04 | 0.05 |
| **Ours:** | | | | | |
| DS-DQN (Vanilla) | 19.2±0.75 | 18.4±2.15 | 19.2±1.17 | 17.2±2.56 | 18.8±1.17 |
| DS-DQN (Radial) | **21.0±0.00** | **21.0±0.00** | **21.0±0.00** | **20.8±0.40** | **19.0±1.41** |
| DS-DQN (SADQN) | **21.0±0.00** | 20.4±1.20 | 19.6±1.50 | 18.4±2.42 | 17.0±3.74 |
| DAS-DQN (Vanilla) | 19.4±1.62 | 19.0±1.41 | 18.8±1.94 | 17.2±1.94 | 13.8±5.88 |
| **SOTA robust agents:** | | | | | |
| RadialDQN | **21.0±0.00** | 19.6±2.80 | −20.2±0.40 | −20.6±0.49 | −21.0±0.00 |
| SADQN | **21.0±0.00** | −19.4±0.80 | −21.0±0.00 | −21.0±0.00 | −21.0±0.00 |
| **CROP small $\sigma$:** | | | | | |
| CROP (Radial) | **21.0±0.00** | 8.6±4.22 | −19.8±0.98 | −21.0±0.00 | −21.0±0.00 |
| CROP (SADQN) | **21.0±0.00** | −19.8±1.47 | −21.0±0.00 | −21.0±0.00 | −21.0±0.00 |
| Freeway | | | | | |
| **Ours:** | | | | | |
| DS-DQN (Vanilla) | 32.6±1.20 | **31.6±1.50** | 30.0±1.10 | 28.0±1.41 | 23.8±1.17 |
| DS-DQN (Radial) | 32.2±0.75 | 31.4±1.02 | **30.4±1.02** | **29.4±0.49** | **30.2±1.47** |
| DS-DQN (SADQN) | 30.0±0.00 | 29.8±0.40 | 27.6±1.50 | 26.8±1.47 | 27.2±1.72 |
| DAS-DQN (Vanilla) | 32.8±1.17 | 30.6±1.36 | 29.0±1.10 | 27.2±0.98 | 26.0±1.26 |
| **SOTA robust agents:** | | | | | |
| RadialDQN | **33.0±1.10** | 29.0±1.70 | 23.4±1.96 | 19.2±1.47 | 20.0±1.10 |
| SADQN | 30.0±0.00 | 27.2±1.17 | 20.4±0.49 | 20.8±0.98 | 18.8±1.33 |
| **CROP small $\sigma$:** | | | | | |
| CROP (Radial) | 32.6±0.49 | 27.4±2.73 | 23.6±1.96 | 20.4±1.62 | 21.2±1.72 |
| CROP (SADQN) | 30.0±0.00 | 27.0±0.89 | 21.4±1.02 | 20.2±0.75 | 19.0±1.41 |
| RoadRunner | | | | | |
| **Ours:** | | | | | |
| DS-DQN (vanilla) | 22300±2771 | 12420±1633 | 1340±2531 | 0±0 | 0±0 |
| DS-DQN (Radial) | 35490±3670 | **33480±3468** | **26580±13025** | 22280±3037 | **19920±1713** |
| DS-DQN (SADQN) | 32280±4712 | 27080±3043 | 21580±2687 | 15660±1861 | 14120±1768 |
| DAS-DQN (Vanilla) | 29300±2534 | 24500±2951 | 21400±2592 | **23800±2733** | 19540±4317 |
| **SOTA robust agents:** | | | | | |
| RadialDQN | **48760±4968** | 26280±2006 | 9700±7188 | 3020±1067 | 760±472 |
| SADQN | 30300±1491 | 2320±1786 | 120±240 | 40±80 | 0±0 |
| **CROP small $\sigma$:** | | | | | |
| CROP (Radial) | 40220±8021 | 24000±3923 | 12220±6516 | 2060±2188 | 940±1169 |
| CROP (SADQN) | 28720±5756 | 1540±714 | 200±276 | 220±286 | 0±0 |

Table 10: The reward of DQN agents under $\ell_2$ PGD attack. The smoothing variance $\sigma$ for our agents is set to $0.1$ in Pong, $0.12$ in Freeway, and $\sigma = 0.1$ in RoadRunner. For all the CROP small $\sigma$ agents, $\sigma = 0.01$.

| Pong | $\ell_2$ PGD attack (used our new attack to evaluate the smoothed agents) | | | | |
|---|---|---|---|---|---|
| $\epsilon(\ell_2)$ | 0.1 | 0.3 | 0.5 | 0.7 | 0.9 |
| **Ours:** | | | | | |
| DS-DQN (Vanilla) | 20.6±0.80 | 15.0±3.10 | 8.2±2.4 | −12.6±4.41 | −16.8±3.70 |
| DS-DQN (Radial) | **21.0±0.00** | 18.8±1.94 | 15.2±3.31 | **9.8±4.26** | **5.2±4.79** |
| DS-DQN (SADQN) | **21.0±0.00** | **20.6±0.49** | **19.8±1.47** | 2.0±12.6 | −16.2±2.64 |
| DAS-DQN (Vanilla) | 19.6±1.02 | 17.4±1.85 | 7.6±2.65 | −10.6±4.63 | −15.0±2.83 |
| **SOTA robust agents:** | | | | | |
| RadialDQN | **21.0±0.00** | 6.0±5.02 | −10.6±12.7 | −19.4±1.20 | −21.0±0.00 |
| SADQN | **21.0±0.00** | −3.6±20.1 | −20.6±0.49 | −20.8±0.40 | −21.0±0.00 |
| **CROP small $\sigma$:** | | | | | |
| CROP (Radial) | **21.0±0.00** | 2.0±3.03 | −21.0±0.00 | −21.0±0.00 | −21.0±0.00 |
| CROP (SADQN) | **21.0±0.00** | −3.6±19.4 | −20.6±0.80 | −20.8±0.40 | −21.0±0.00 |
| Freeway | | | | | |
| **Ours:** | | | | | |
| DS-DQN (Vanilla) | 32.6±1.02 | **31.4±1.36** | **30.0±1.55** | 29.0±1.10 | 26.8±1.17 |
| DS-DQN (Radial) | 31.8±0.98 | 31.2±1.17 | **30.0±2.10** | 28.0±1.67 | 26.6±1.36 |
| DS-DQN (SADQN) | 30.0±0.00 | 29.4±0.80 | 28.2±1.17 | 27.4±1.20 | 27.2±0.75 |
| DAS-DQN (Vanilla) | 32.6±1.02 | 31.0±1.41 | 29.8±0.75 | **29.2±1.94** | **27.6±2.06** |
| **SOTA robust agents:** | | | | | |
| RadialDQN | **33.0±1.10** | 29.2±0.98 | 23.8±1.47 | 24.8±2.79 | 22.4±0.80 |
| SADQN | 30.0±0.63 | 26.4±4.03 | 27.4±3.14 | 27.2±2.32 | **27.6±2.06** |
| **CROP small $\sigma$:** | | | | | |
| CROP (Radial) | 32.6±0.49 | 28.8±1.83 | 23.8±0.98 | 23.8±1.33 | 23.2±0.98 |
| CROP (SADQN) | 30.0±0.63 | 25.6±3.14 | 26.2±1.83 | 27.2±2.32 | 27.2±2.32 |
| RoadRunner | | | | | |
| **Ours:** | | | | | |
| DS-DQN (Vanilla) | 31560±5942 | 24520±2986 | 21560±2585 | 17500±4693 | 13780±3396 |
| DS-DQN (Radial) | 32480±6251 | 23200±2700 | 22460±2922 | **20120±1212** | **21280±1251** |
| DS-DQN (SADQN) | 30900±5055 | 21320±2785 | 19700±1485 | 19580±1206 | 17240±1453 |
| DAS-DQN (Vanilla) | 27880±3626 | 22440±2526 | **24860±3197** | 19380±1488 | 18940±1904 |
| **SOTA robust agents:** | | | | | |
| RadialDQN | **45720±11105** | **34480±3292** | 19080±4370 | 10440±4340 | 8300±2663 |
| SADQN | 38060±5522 | 18860±3364 | 6260±3212 | 3280±1785 | 1300±1023 |
| **CROP small $\sigma$:** | | | | | |
| CROP (Radial) | 40240±8007 | 27020±11677 | 17980±5887 | 8000±3592 | 6120±1957 |
| CROP (SADQN) | 40900±7677 | 20440±1678 | 5960±4273 | 2540±1263 | 1100±593 |

Table 11 shows the reward of PPO agents under different $\ell_\infty$ attacks. Note that we trained each agent 15 times and reported the median of the performance as suggested in Zhang et al. (2020) to get a fair and comparable result. We also used our new attack in Section 3.2 to evaluate the smoothed agents. Our AS-PPO exhibits a better trade-off between the clean reward and the robust reward under attacks in all environments and performs especially well in Walker. Our AS-PPO also receives a much higher clean reward in Humanoid. Humanoid has a high state space dimension (376) and is harder to train than Walker and Hopper. This result suggests that our approach can further help agents learn in the non-adversarial setting.

Table 11: The reward of PPO agents under different attacks. The smoothing variance $\sigma$ for all the smoothed agents is set to 0.2 in Walker, 0.3 in Hopper, and 0.4 in Humanoid. The $\ell_\infty$ attack budget is set to 0.05 in Walker and 0.075 in Hopper and Humanoid.

| Walker | Clean Reward | Reward under Attack | | | |
| --- | --- | --- | --- | --- | --- |
| | | Random | Critic | MAD | Min-RS |
| **Ours:** | | | | | |
| AS-PPO | $\mathbf{4969.3 \pm 137.1}$ | $\mathbf{5039.4 \pm 1132.8}$ | $\mathbf{5488.0 \pm 568.0}$ | $\mathbf{5290.4 \pm 966.9}$ | $\mathbf{4322.8}$ |
| **SAPPO:** | | | | | |
| SGLD | $4911.8 \pm 188.9$ | $4894.8 \pm 139.9$ | $5019.0 \pm 65.2$ | $4755.7 \pm 413.1$ | $2605.6$ |
| Convex | $4486.6 \pm 60.7$ | $4475.0 \pm 48.7$ | $4572.0 \pm 52.3$ | $4343.4 \pm 329.4$ | $2168.2$ |
| **SAPPO+RS:** | | | | | |
| SGLD+RS | $4893.6 \pm 220.2$ | $4876.4 \pm 112.7$ | $5015.8 \pm 74.6$ | $4782.2 \pm 290.7$ | $2615.3$ |
| Convex+RS | $4475.9 \pm 41.1$ | $4467.7 \pm 41.5$ | $4573.7 \pm 50.6$ | $4369.9 \pm 366.4$ | $2794.1$ |
| Hopper | | | | | |
| **Ours:** | | | | | |
| AS-PPO | $3666.9 \pm 283.4$ | $3545.5 \pm 373.1$ | $\mathbf{3705.8 \pm 4.8}$ | $2916.4 \pm 874.3$ | $1557.9$ |
| **SAPPO:** | | | | | |
| SGLD | $3523.1 \pm 329.0$ | $3080.2 \pm 745.4$ | $3665.5 \pm 8.2$ | $2996.6 \pm 786.4$ | $1403.3$ |
| Convex | $\mathbf{3704.1 \pm 2.2}$ | $\mathbf{3708.7 \pm 23.8}$ | $3698.4 \pm 4.4$ | $3443.1 \pm 466.7$ | $1235.8$ |
| **SAPPO+RS:** | | | | | |
| SGLD+RS | $2689.6 \pm 793.5$ | $2739.9 \pm 715.1$ | $3667.5 \pm 7.4$ | $2809.9 \pm 787.4$ | $\mathbf{1560.9}$ |
| Convex+RS | $3685.2 \pm 15.1$ | $3683.8 \pm 37.4$ | $3683.5 \pm 9.4$ | $\mathbf{3611.0 \pm 150.9}$ | $1529.2$ |
| Humanoid | | | | | |
| **Ours:** | | | | | |
| AS-PPO | $\mathbf{6977.9 \pm 47.7}$ | $\mathbf{6960.4 \pm 27.2}$ | $\mathbf{7013.2 \pm 25.4}$ | $6825.3 \pm 953.2$ | $6070.3$ |
| **SAPPO:** | | | | | |
| SGLD | $6624.0 \pm 25.5$ | $6614.1 \pm 21.4$ | $6587.0 \pm 23.1$ | $6586.4 \pm 23.5$ | $6200.5$ |
| Convex | $6400.6 \pm 156.8$ | $6207.9 \pm 783.3$ | $6397.9 \pm 35.6$ | $6379.5 \pm 30.5$ | $4707.2$ |
| **SAPPO+RS:** | | | | | |
| SGLD+RS | $6968.1 \pm 19.0$ | $6954.6 \pm 19.1$ | $6927.6 \pm 26.2$ | $\mathbf{6883.8 \pm 35.5}$ | $\mathbf{6657.9}$ |
| Convex+RS | $6517.2 \pm 56.1$ | $6524.0 \pm 41.9$ | $6520.2 \pm 42.2$ | $6472.2 \pm 61.2$ | $3946.7$ |

## A.16 TRAINING VARIANCE OF DS-DQN

We also reported the training variance of our DS-DQN (Vanilla) in Table 12. Unlike AS-PPO, DS-DQN has a much smaller variance and hence we only show the results of one training of DS-DQN in our main experiments.

Table 12: The training variance of DS-DQN (Vanilla). The L-inf attack budget of robust reward is set to 0.03 in Pong and Freeway, and 0.01 in RoadRunner. It can be seen that the training variance of DS-DQN (Vanilla) is small and the median of these 5 agents is very close to what we reported in Table 2 and Figure 2.

| Pong | Agent 1 | Agent 2 | Agent 3 | Agent 4 | Agent 5 | AVG & STD |
|---|---|---|---|---|---|---|
| Clean Reward | 21.0 | 21.0 | 20.8 | 20.8 | 20.4 | $20.8 \pm 0.24$ |
| Certified Radius | 0.0614 | 0.0556 | 0.0629 | 0.0617 | 0.0546 | $0.0592 \pm 0.0038$ |
| Robust Reward | 19.0 | 19.8 | 19.6 | 20.2 | 18.4 | $19.4 \pm 0.71$ |
| Freeway | | | | | | |
| Clean Reward | 32.8 | 32.4 | 32.0 | 32.6 | 32.0 | $32.4 \pm 0.36$ |
| Certified Radius | 0.0891 | 0.0886 | 0.0951 | 0.0891 | 0.0914 | $0.0907 \pm 0.0027$ |
| Robust Reward | 30.2 | 28.4 | 27.4 | 29.6 | 24.4 | $28.0 \pm 2.28$ |
| RoadRunner | | | | | | |
| Clean Reward | 36820 | 30120 | 38900 | 30400 | 36820 | $34612 \pm 4064$ |
| Certified Radius | 0.0739 | 0.0769 | 0.0635 | 0.0648 | 0.0769 | $0.0712 \pm 0.0066$ |
| Robust Reward | 20060 | 22360 | 21820 | 25380 | 19340 | $21792 \pm 2356$ |

## A.17 OTHER EXPERIMENT RESULTS

Table 13 shows the results of AS-PPO trained under difference smoothing variance $\sigma$. In Walker, we found that $\sigma = 0.2$ is the best setting among all the other values. In Hopper, there is a trade-off between the clean reward and the robust reward under different attacks. This tendency also appears in Humanoid. We choose the $\sigma$ that achieves the highest reward under the Min-RS attack as our setting.

Table 13: The performance of AS-PPO with different smoothing variance $\sigma$.

| Walker $\epsilon(\ell_\infty) = 0.05$ | | | |
|---|---|---|---|
| Smoothing Variance $\sigma$ | 0.1 | 0.2 | 0.3 |
| Clean | $4938.82 \pm 16.92$ | $\mathbf{4969.28 \pm 137.07}$ | $4804.48 \pm 582.28$ |
| Random | $4921.7 \pm 23.59$ | $\mathbf{5039.42 \pm 1132.80}$ | $4824.78 \pm 101.88$ |
| Critic | $5006.05 \pm 44.92$ | $\mathbf{5487.99 \pm 567.96}$ | $4923.6 \pm 1116.18$ |
| MAD | $4933.05 \pm 81.04$ | $\mathbf{5290.36 \pm 966.92}$ | $4665.54 \pm 1244.06$ |
| Min-RS | 3203.09 | $\mathbf{4322.79}$ | 3339.11 |
| Hopper $\epsilon(\ell_\infty) = 0.075$ | | | |
| Smoothing Variance $\sigma$ | 0.1 | 0.2 | 0.3 |
| Clean | $3650.53 \pm 14.50$ | $\mathbf{3758.52 \pm 44.62}$ | $3666.92 \pm 283.36$ |
| Random | $3627.58 \pm 24.61$ | $\mathbf{3719.45 \pm 26.23}$ | $3545.54 \pm 373.12$ |
| Critic | $3606.85 \pm 29.53$ | $\mathbf{3758.99 \pm 12.26}$ | $3705.8 \pm 4.76$ |
| MAD | $2763.29 \pm 772.89$ | $\mathbf{3360.12 \pm 520.56}$ | $2916.44 \pm 874.28$ |
| Min-RS | 1010.31 | 1354.46 | $\mathbf{1557.93}$ |
| Humanoid $\epsilon(\ell_\infty) = 0.075$ | | | |
| Smoothing Variance $\sigma$ | 0.3 | 0.4 | 0.5 |
| Clean | $\mathbf{7157.50 \pm 463.39}$ | $6977.93 \pm 47.74$ | $6847.94 \pm 28.56$ |
| Random | $\mathbf{7135.64 \pm 22.92}$ | $6960.41 \pm 27.16$ | $6719.15 \pm 896.53$ |
| Critic | $\mathbf{7277.57 \pm 35.71}$ | $7013.24 \pm 25.35$ | $6853.16 \pm 21.31$ |
| MAD | $\mathbf{6880.31 \pm 969.87}$ | $6825.31 \pm 953.20$ | $6762.92 \pm 1007.46$ |
| Min-RS | 4764.57 | $\mathbf{6070.29}$ | 5577.89 |

