# OpenReview forum: "Tactics of Robust Deep Reinforcement Learning with Randomized Smoothing"
_ICLR.cc/2024/Conference — Submitted to ICLR 2024_

### Official Review · Reviewer_ZBr8 · 2023-10-29

**Soundness:** 3 good
**Presentation:** 3 good
**Contribution:** 2 fair
**Rating:** 5
**Confidence:** 3

**Summary:**

This paper focuses on CROP and an updated reward evaluation and defense metrics.

**Strengths:**

- This paper is clearly written and easy to follow.
- This paper targets a ‘mis’-evaluation in a past work: CROP, which is a clear motivation.

**Weaknesses:**

- This work targets only on ONE random smoothing paper: CROP, which limits this paper’s signification. There is another concurrent work on random smoothing [1] with CROP. No discussion and comparison about [1] makes the contribution of this work unclear.
- This main concern the reviewer have is that this work is completely based on CROP and is a correction of CROP. Whether this paper should be published as a full research paper in ICLR main track is doubtful.

[1]. Policy smoothing for provably robust reinforcement learning. ICLR 2022

**Questions:**

N/A

---

> ### Author Response · Authors · 2023-11-19
> **Author Response**
>
> Thank you for pointing out your main concern!
>
> >**Q1:** This work targets only on ONE random smoothing paper: CROP, which limits this paper’s signification. There is another concurrent work on random smoothing [1] with CROP. No discussion and comparison about [1] makes the contribution of this work unclear.
>
> **A1:** In [1], they focused on the issue that the bounds in the Supervised Learning setting cannot directly transfer to the RL setting because of the non-static nature of RL. They provided another proof for the bounds and pointed out that the proof in CROP might have some issues. However, [1] also uses the same naive smoothing during inference, which will lead to the same problem as CROP did. We only compare to CROP since the environments we use are closer to CROP. There will be no difference between CROP and [1] in terms of the practical implementation. Hence, [1] still falls into the issue of low clean reward when the smoothing variance increases. We consider this issue important since the agents with strong guarantees but poorly performed are unacceptable.
>
> [1] Kumar, et al. “Policy Smoothing for Provably Robust Reinforcement Learning”, ICLR 2022
>
> **#Additional experiments performed in rebuttal**
>
> We would like to highlight the additional experiments we performed during the rebuttal. We compare our methods with recent methods including ATLA [2] and WocaR [3]. The DQN results are shown in Table A.1 and A.2. The PPO results are shown in Table B.1 and B.2. Our methods still outperform these methods in most of the settings, which strengthens our contributions.
>
> **Table A.1.** The reward of DS-DQN and WocaR-DQN under PGD attack in the Pong environment
> (Pong) eps|0.01|0.02|0.03|0.04|0.05
> ---|---|---|---|---|---
> Our DSDQN (Vanilla)|19.2+-0.75|**18.4+-2.15**|**19.2+-1.17**|**17.2+-2.56**|**18.8+-1.17**
> WocaRDQN|**21.0+-0.0**|-21.0+-0.0|-21.0+-0.0|-21.0+-0.0|-21.0+-0.0
>
> **Table A.2.** The reward of DS-DQN and WocaR-DQN under PGD attack in the Freeway environment
> (Freeway) eps|0.01|0.02|0.03|0.04|0.05
> ---|---|---|---|---|---
> Our DSDQN (Vanilla)|**32.6+-1.20**|**31.6+-1.50**|**30.0+-1.10**|**28.0+-1.41**|**23.8+-1.17**
> WocaRDQN|21.4 +- 1.36|21.8 +- 1.17|21.0 +- 0.89|21.0 +- 0.63|21.0 +- 1.10
>
> **Table B.1.** The reward of AS-PPO, Radial-PPO, ATLA-PPO, and WocaR-PPO under attacks in the Walker environment
> (Walker) eps=0.05|clean|random|critic|MAD|Min RS|Optimal attack
> ---|---|---|---|---|---|---
> AS-PPO (Ours)|4969|5039|**5488**|**5290**|**4323**|**4296**
> Radial-PPO|**5251**|**5184**|No result|4494|3572|3320
> ATLA-PPO|3920|3779|3915|3963|3219|3463
> WocaR-PPO|4156|4244|No result|4177|4093|3770
>
> **Table B.2.** The reward of AS-PPO, Radial-PPO, ATLA-PPO, and WocaR-PPO under attacks in the Hopper environment
> (Hopper) eps=0.075|clean|random|critic|MAD|Min RS|Optimal attack
> ---|---|---|---|---|---|---
> AS-PPO (Ours)|3667|3546|**3706**|2916|1558|1500
> Radial-PPO|**3740**|**3729**|No result|3214|2141|1722
> ATLA-PPO|3487|3474|3524|3081|1567|1224
> WocaR-PPO|3616|3633|No result|**3541**|**3277**|**2390**
>
> [2] Zhang, et al. “Robust Reinforcement Learning on State Observations with Learned Optimal Adversary”, ICLR 2021
>
> [3] Liang, et al. “Efficient Adversarial Training without Attacking: Worst-Case-Aware Robust Reinforcement Learning”, NeurIPS 2022
>
> **Summary**
>
> In summary, we have:
> * In Q1, explain that the issue in CROP actually happens generally and the paper [1] also has the same issue.
> * Provide additional experiments and show that our methods still outperform recent methods in most of the settings.
>
> We believe that we have addressed all your concerns. Please let us know if you still have any reservations and we would be happy to address them!

---

> ### Author Response · Authors · 2023-11-21
> **Request Rebuttal feedback from Reviewer ZBr8**
>
> Dear Reviewer ZBr8,
>
> We believe that we have addressed all your concerns. We also provide additional experiments in Table A.1, A.2, B.1, and B.2. Please let us know if you still have any reservations and we would be happy to address them. Thank you!

---

> > ### Comment · Reviewer_ZBr8 · 2023-11-22
> > **Reply**
> >
> > Thank you for your response. After reviewing the response, it's still unclear to me how to position the contribution of this paper. Thus, I keep my score.

---

> > > ### Author Response · Authors · 2023-11-23
> > > **Further clarification on our contributions and position of this paper**
> > >
> > > Thanks for the reply and sharing your concerns! Sorry for making you feel unclear about our contribution. We feel that the current issue of the CROP framework is significant since a smoothed agent with a high guarantee would still be meaningless given that the clean reward is already largely degraded. Our agents are also the first SOTA with a strong provable guarantee and we believe that our contribution is not minor, which can open a new direction of further strengthening the robustness guarantee for future works.
> > >
> > > Here we give a detailed list of our 4 major contributions to this work to make the understanding easier:
> > >
> > > 1. **Contributions related to the simple Randomized Smoothing (RS):**
> > >     * We identify the failure mode of the existing smoothed DRL agents, showing that the naively smoothed agents have poor trade-offs on the clean reward and robustness.
> > >    * We point out that the robustness of the CROP agents might be overestimated due to the smoothing strategy and attack they used. We fix this issue by introducing hard RS and a stronger attack.
> > >    * We extend CROP’s proposed robust guarantee for DQN agents to the PPO setting and defined action bound. To our best knowledge, the action bound for PPO has never been derived before. We also do experiments on this bound (see Appendix A.13).
> > > 2. **Contributions related to robust DRL agents:**
> > >     * Our agents achieved state-of-the-art results under attack. Following the reviewers’ suggestions, We include new baselines and show that our agent still outperforms the current best WocaR-RL in most of the environments.
> > >     * Our agent is the first state-of-the-art agent with a high robustness guarantee at the same time (certified radius and reward lower bound), while the previous state-of-the-art only evaluated their agents under empirical attack.
> > > 3. **Contributions related to Randomized Smoothing (RS) in DRL:**
> > >     * We develop the first robust DRL training algorithms leveraging randomized smoothing for both discrete actions (DS-DQN) and continuous actions (AS-PPO).
> > >     * We show that simply training with RS does not work, and is necessary to use denoised smoothing and adversarial training.
> > >     * We point out that different smoothing strategies can affect the robustness guarantee. (i.e. Our hard RS strategy is not affected by the output range of the Q-network and generally achieves a larger certified radius.)
> > > 4. **Contributions related to adversarial attack:**
> > >     * We develop a new attack aiming at attacking smoothed agents. This attack model we proposed can be easily used on any attack based on optimization. We show how our setting can be applied to the PA-AD attack, denoted as S-PA-AD, and lead to a stronger attack for smoothed agents.
> > >
> > > Thanks for your valuable feedback! Please let us know if you have further concerns. We are happy to address them!

---

### Official Review · Reviewer_fZcx · 2023-10-29

**Soundness:** 3 good
**Presentation:** 2 fair
**Contribution:** 2 fair
**Rating:** 5
**Confidence:** 3

**Summary:**

This paper proposes a smoothing method to find a trade-off between the robustness of the agent under adversarial attack and reward performance. In particular, it introduces two smoothed agents: DS-DQN for discrete actions and AS-PPO for continuous actions. The method is evaluated under different attack models and is compared with a previous method, CROP, and other baselines on three Atari games and three continuous control tasks.

**Strengths:**

- This paper tackles an important problem of learning robust policy in deep reinforcement learning setup.
- The empirical results seem better compared to the baseline on the tested environment.
- The methods appear to be easy to implement on top of existing algorithms (DQN, PPO).

**Weaknesses:**

- The proposed method and its presentation rely heavily on the existing CROP method, thereby limiting its novelty.
- The method's description is dispersed throughout the paper, complicating comprehension, and the approaches for discrete and continuous cases appear to differ.
- The implications of robust certification remain unclear within the context of the evaluated task.

**Questions:**

What are the implications of a robust certificate in this context? How is it computed for experimental evaluation, such as for Atari Pong?
In DS-DQN, a random number generated from a Gaussian distribution is added to the state during training. A denoiser network is then employed to reconstruct the original state. Does this imply that, in an ideal scenario, the denoiser network essentially mirrors the Gaussian distribution introduced to the state initially? If so, what is the purpose of training such a denoiser when we already know the noise model added to the input? How does the denoiser contribute to achieving better rewards and enhanced robustness?

Is a denoiser model used for the continuous (AS-PPO) case? What constitutes the smoothing component in this PPO scenario? What does \Delta S_t represent in the paragraph following Equation 9?

I am not clear on why this denoiser method helps in achieving a better trade-off between robustness and reward. A detailed description would assist in understanding the empirical performance.

Are the attack models for discrete (DQN) and continuous (PPO) cases the same?

---

> ### Author Response · Authors · 2023-11-19
> **Author Response (1/2)**
>
> Thanks for pointing out your main concerns!
>
> >**Q1:** The proposed method and its presentation rely heavily on the existing CROP method, thereby limiting its novelty.
>
> **A1:** There is another paper [1] that also studies randomized smoothing in RL. This paper and CROP both do randomized smoothing during the inference, which leads to the problem we discussed. Although we use the CROP paper to identify the problem, this is actually a general issue that happens when introducing randomized smoothing naively.  We consider this issue important since the agents with strong guarantees but poorly performed are unacceptable.
>
> [1] Kumar, et al. “Policy Smoothing for Provably Robust Reinforcement Learning”, ICLR 2022
>
> >**Q2:** The method's description is dispersed throughout the paper, complicating comprehension, and the approaches for discrete and continuous cases appear to differ.
>
> **A2:** Sorry for making it difficult for you to read. Our proposed method of robust DRL agents for both discrete and continuous actions (DS-DQN in sec 4.1 and AS-PPO in sec 4.2) is structured as follows:
> 1. **Motivating example:** We first point out where the previous works did not work well. Hence, we try to leverage RS in the training and design S-DQN and S-PPO. However, we found that there are many problems with this simple design. That's why we design DS-DQN and AS-PPO.
> 2. **Training:** This part provides the details of our methods. We also include flow charts in Appendix A.2 and A.3 to give a whole picture of our methods.
> 3. **Testing:** The most important thing here is to show our smoothing strategies. We use hard RS for DS-DQN and median smoothing for AS-PPO since there are some issues with the smoothing strategy in CROP.
> 4. **Attack:** Since all our attacks are based on our new attack, we especially introduce how we evaluate the robust reward of our agents.
>
> We believe structuring the main methods in this order can best describe the process of how we discover a problem, how we solve the problem, and how we evaluate our algorithms. As for the connection between the discrete case and the continuous case, they both show how to effectively leverage randomized smoothing to train a smoothed agent that is provably robust. We feel that it is necessary to show that no matter in the discrete or continuous setting, our framework achieves high clean reward, high robust reward, and strong guarantees simultaneously.
>
> >**Q3:** What are the implications of a robust certificate in this context? How is it computed for experimental evaluation, such as for Atari Pong? In DS-DQN, a random number generated from a Gaussian distribution is added to the state during training. A denoiser network is then employed to reconstruct the original state. Does this imply that, in an ideal scenario, the denoiser network essentially mirrors the Gaussian distribution introduced to the state initially? If so, what is the purpose of training such a denoiser when we already know the noise model added to the input? How does the denoiser contribute to achieving better rewards and enhanced robustness?
>
> **A3:** The robust certificate means the certified radius and reward lower bound for DQN and the reward lower bound for PPO. We provide the details of how we estimate and compute those bounds in practice in Appendix A9.  As for the DS-DQN, simply removing the noise that is already known cannot improve the robustness. The denoiser should be viewed as a part of the agent, which does not necessarily denoise the state to the original one since we also use the temporal difference loss. The denoiser makes the agents more tolerant against the Gaussian noise and hence the problem of decreasing clean reward can be mitigated. Since DS-DQN is more tolerant against the noise, a much larger smoothing variance is applied to enhance the robustness.
>
> >**Q4:** Is a denoiser model used for the continuous (AS-PPO) case? What constitutes the smoothing component in this PPO scenario? What does \Delta S_t represent in the paragraph following Equation 9?
>
> **A4:** We did not use the denoiser for the PPO case since the input state is not an image. PPO agents are much more tolerant to the Gaussian noise and allow us to directly train the agent with randomized smoothing. The $\Delta S_t$ is the perturbation from the adversary that we use while training AS-PPO. In the stage of optimizing the adversary, we intend to find a set of perturbations  ${\Delta S_1, …, \Delta S_T}$ that minimizes the cumulative reward.

---

> ### Author Response · Authors · 2023-11-19
> **Author Response (2/2)**
>
> >**Q5:** I am not clear on why this denoiser method helps in achieving a better trade-off between robustness and reward. A detailed description would assist in understanding the empirical performance.
>
> **A5:** The naively smoothing strategy in CROP decreases the clean reward since the agents are not trained to tolerate the large Gaussian noise. The larger the smoothing variance is, the more the clean reward decreases. We introduce a denoiser to make agents more tolerant to the Gaussian noise. Therefore, we can make the smoothing variance larger, which enhances the robustness guarantee, without reducing the clean reward.
>
> >**Q6:** Are the attack models for discrete (DQN) and continuous (PPO) cases the same?
>
> **A6:** There are all L-p norm attacks, but the PPO case is more complicated since there is no Q-value to reduce. For the PPO attacks, please refer to [2] for more details. Our PPO attacks are based on their methods.
>
> [2] Zhang, et al. "Robust Deep Reinforcement Learning against Adversarial Perturbations on State Observations", NeurIPS 2020
>
> **#Additional experiments performed in rebuttal**
>
> We would like to highlight the additional experiments we performed during the rebuttal. We compare our methods with recent methods including ATLA [3] and WocaR [4]. The DQN results are shown in Table A.1 and A.2. The PPO results are shown in Table B.1 and B.2. Our methods still outperform these methods in most of the settings, which strengthens our contributions.
>
> **Table A.1.** The reward of DS-DQN and WocaR-DQN under PGD attack in the Pong environment
> (Pong) eps|0.01|0.02|0.03|0.04|0.05
> ---|---|---|---|---|---
> Our DSDQN (Vanilla)|19.2+-0.75|**18.4+-2.15**|**19.2+-1.17**|**17.2+-2.56**|**18.8+-1.17**
> WocaRDQN|**21.0+-0.0**|-21.0+-0.0|-21.0+-0.0|-21.0+-0.0|-21.0+-0.0
>
> **Table A.2.** The reward of DS-DQN and WocaR-DQN under PGD attack in the Freeway environment
> (Freeway) eps|0.01|0.02|0.03|0.04|0.05
> ---|---|---|---|---|---
> Our DSDQN (Vanilla)|**32.6+-1.20**|**31.6+-1.50**|**30.0+-1.10**|**28.0+-1.41**|**23.8+-1.17**
> WocaRDQN|21.4 +- 1.36|21.8 +- 1.17|21.0 +- 0.89|21.0 +- 0.63|21.0 +- 1.10
>
> **Table B.1.** The reward of AS-PPO, Radial-PPO, ATLA-PPO, and WocaR-PPO under attacks in the Walker environment
> (Walker) eps=0.05|clean|random|critic|MAD|Min RS|Optimal attack
> ---|---|---|---|---|---|---
> AS-PPO (Ours)|4969|5039|**5488**|**5290**|**4323**|**4296**
> Radial-PPO|**5251**|**5184**|No result|4494|3572|3320
> ATLA-PPO|3920|3779|3915|3963|3219|3463
> WocaR-PPO|4156|4244|No result|4177|4093|3770
>
> **Table B.2.** The reward of AS-PPO, Radial-PPO, ATLA-PPO, and WocaR-PPO under attacks in the Hopper environment
> (Hopper) eps=0.075|clean|random|critic|MAD|Min RS|Optimal attack
> ---|---|---|---|---|---|---
> AS-PPO (Ours)|3667|3546|**3706**|2916|1558|1500
> Radial-PPO|**3740**|**3729**|No result|3214|2141|1722
> ATLA-PPO|3487|3474|3524|3081|1567|1224
> WocaR-PPO|3616|3633|No result|**3541**|**3277**|**2390**
>
> [3] Zhang, et al. “Robust Reinforcement Learning on State Observations with Learned Optimal Adversary”, ICLR 2021
>
> [4] Liang, et al. “Efficient Adversarial Training without Attacking: Worst-Case-Aware Robust Reinforcement Learning”, NeurIPS 2022
>
> **Summary**
>
> In summary, we have:
> * In Q1, explain that the issue in CROP also happens in other papers using RS for RL.
> * In Q2, explain the structure of our writing and show the connection between the discrete and the continuous parts.
> * In Q3 and Q5, explain how the denoiser works and why denoised smoothing can lead to better robustness.
> * In Q4, Clarify the settings and the notations of AS-PPO.
> * In Q6, explain that the attacks for DQN and PPO cases are different.
> * Provide additional experiments and show that our methods still outperform recent methods in most of the settings.
>
> We believe that we have addressed all your concerns. Please let us know if you still have any reservations and we would be happy to address them!

---

> ### Author Response · Authors · 2023-11-21
> **Request Rebuttal feedback from Reviewer fZcx**
>
> Dear Reviewer fZcx,
>
> We believe that we have addressed all your concerns through Q1~Q6. We also provide additional experiments in Table A.1, A.2, B.1, and B.2. Please let us know if you still have any reservations and we would be happy to address them. Thank you!

---

### Official Review · Reviewer_RhGb · 2023-10-30

**Soundness:** 2 fair
**Presentation:** 2 fair
**Contribution:** 2 fair
**Rating:** 5
**Confidence:** 4

**Summary:**

This paper tackles the challenge of enhancing the robustness of deep reinforcement learning (DRL) agents while preserving their utility. Although randomized smoothing offers robustness guarantees, its direct implementation in the DRL context often results in a trade-off between utility and robustness. To overcome this limitation, the authors introduce two innovative algorithms, DS-DQN and AS-PPO, tailored to train DRL agents that achieve high clean rewards and robustness certification in both discrete and continuous action spaces. DS-DQN and AS-PPO surpass previous robust DRL agents and introduce a more potent adversarial attack. Their contributions encompass addressing issues in prior methods, extending robustness guarantees to PPO settings, and introducing action bounds for continuous-action agents.

**Strengths:**

1. The paper's motivation, stemming from the shortcomings of existing smoothed agents and the trade-off between robustness and performance, is intriguing. The introduction of a robust guarantee sets this paper apart from mere proposals of simple robust RL training methods. It offers a deep understanding of RL robustness within the context of random smoothing.

2. The methods presented in this paper exhibit versatility by working across various types of tasks. The authors make commendable efforts to demonstrate empirical contributions on different domains, showcasing not only robust performance under empirical attacks but also robustness guarantees.

**Weaknesses:**

1. It would enhance clarity to use pseudocode diagrams to illustrate the algorithms' flow. Visual aids can make the presentation of the algorithms more accessible.

2. The experiment section suffers from suboptimal writing and presentation. The experimental results lack the strength of evidence needed to robustly support the claims made, and addressing this issue could significantly improve the paper's overall quality.

3. The paper falls short in the discussion of limitations. A more comprehensive exploration of potential limitations would provide a well-rounded view.

**Questions:**

1. It's disappointing that the paper does not include ATLA[1] and WocaR-RL[2] as robust baselines. While the selected baselines do provide a reasonable basis for comparison, it is necessary and valuable to discuss these adversarial robust RL papers (as well as others that are not mentioned) in the related work section. I am concerned that the literature survey of robust RL baselines by the authors might not be comprehensive, especially with regard to recent works. (The authors do cite [1], but why do not discuss ATLA?)

2. Furthermore, I find the presentation of the experiments to be somewhat confusing. In Table 3, why is the comparison limited to SA-PPO and vanilla PPO, and where are Radial-PPO and other PPO-based baselines? It appears that the proposed method performs better only on larger attack budgets. However, in Table 8, it shows that AS-PPO does not have a significant advantage, or even any advantage, on large budgets.

3. In Table 5, the paper fails to clearly demonstrate the advantages of DS-DQN and AS-PPO; the terms "high" or "highest" are quite vague and do not provide an intuitive representation of contributions.

4. This paper introduces attack methods based on RS. However, the authors do not compare their attack method with the strongest evasion attacks, such as PA-AD[3], nor do they provide a discussion of prior attack methods in the related work.

[1]Robust Reinforcement Learning on State Observations with Learned Optimal Adversary. Huan Zhang, Hongge Chen, Duane Boning, Cho-Jui Hsieh. ICLR 2021.

[2]Efficient Adversarial Training without Attacking: Worst-Case-Aware Robust Reinforcement Learning. Yongyuan Liang, Yanchao Sun, Ruijie Zheng, Furong Huang. Neurips 2022.

[3]Who Is the Strongest Enemy? Towards Optimal and Efficient Evasion Attacks in Deep RL. Yanchao Sun, Ruijie Zheng, Yongyuan Liang, Furong Huang. ICLR 2022.

---

> ### Author Response · Authors · 2023-11-19
> **Author Response (1/2)**
>
> Thank you for your detailed and valuable feedback!
>
> >**Q1:** It would enhance clarity to use pseudocode diagrams to illustrate the algorithms' flow. Visual aids can make the presentation of the algorithms more accessible.
>
> **A1:** Please refer to the block diagram in Appendix A2 and A3, which gives a general idea of how the algorithms work.
>
> >**Q2:** The paper falls short in the discussion of limitations. A more comprehensive exploration of potential limitations would provide a well-rounded view.
>
> **A2:** A limitation of the guarantees can be the curse of dimensionality. Randomized smoothing provides a guarantee under L-2 norm bounded attack, which can generalize to any L-p norm attack. However, in the high dimensional space, there will be very small or no guarantee for L-inf attacks, because the guarantee will shrink by a factor of $\sqrt{d}$ where $d$ is the dimension.
>
> >**Q3:** It's disappointing that the paper does not include ATLA[1] and WocaR-RL[2] as robust baselines
>
> **A3:** Thank you for your suggestions! Following your suggestions, we have conducted additional experiments to include more baseline of Radial-RL, ATLA, and WocaR-RL in Tables A.1, A.2,  B.1, and B.2.
>
> The DQN results are shown in Table A.1 and A.2. We compare our DS-DQN (Vanilla), which only leverages denoised smoothing without further using a robust base model, with WocaR-DQN. The ATLA algorithm can only apply to the PPO setting, and hence no need to include it here. WocaR-DQN performs well only under a small attack budget and is not comparable to our DS-DQN (Vanilla). Although we use a much larger budget to evaluate all the DQN agents. we feel that this is necessary since previous works such as SA-DQN or Radial-DQN already demonstrated strong robustness under a small attack budget.
>
> The PPO results are shown in Table B.1 and B.2. We do an additional comparison with Radial-PPO, ATLA-PPO, and WocaR-PPO in Walker and Hopper environments. Our AS-PPO outperforms these baselines in the Walker environment, while WocaR-PPO and Radial-PPO perform better in the Hopper environment.
>
> Nevertheless, We would like to highlight that all these baselines do not come with robustness guarantees like our agents. Our agents are not only empirically robust but also provably robust. It is especially hard to make the provably robust agents also perform well under empirical attack.
>
> **Table A.1.** The reward of DS-DQN and WocaR-DQN under PGD attack in the Pong environment
> (Pong) eps|0.01|0.02|0.03|0.04|0.05
> ---|---|---|---|---|---
> Our DSDQN (Vanilla)|19.2+-0.75|**18.4+-2.15**|**19.2+-1.17**|**17.2+-2.56**|**18.8+-1.17**
> WocaRDQN|**21.0+-0.0**|-21.0+-0.0|-21.0+-0.0|-21.0+-0.0|-21.0+-0.0
>
> **Table A.2.** The reward of DS-DQN and WocaR-DQN under PGD attack in the Freeway environment
> (Freeway) eps|0.01|0.02|0.03|0.04|0.05
> ---|---|---|---|---|---
> Our DSDQN (Vanilla)|**32.6+-1.20**|**31.6+-1.50**|**30.0+-1.10**|**28.0+-1.41**|**23.8+-1.17**
> WocaRDQN|21.4 +- 1.36|21.8 +- 1.17|21.0 +- 0.89|21.0 +- 0.63|21.0 +- 1.10
>
> **Table B.1.** The reward of AS-PPO, Radial-PPO, ATLA-PPO, and WocaR-PPO under attacks in the Walker environment
> (Walker) eps=0.05|clean|random|critic|MAD|Min RS|Optimal attack
> ---|---|---|---|---|---|---
> AS-PPO (Ours)|4969|5039|**5488**|**5290**|**4323**|**4296**
> Radial-PPO|**5251**|**5184**|No result|4494|3572|3320
> ATLA-PPO|3920|3779|3915|3963|3219|3463
> WocaR-PPO|4156|4244|No result|4177|4093|3770
>
> **Table B.2.** The reward of AS-PPO, Radial-PPO, ATLA-PPO, and WocaR-PPO under attacks in the Hopper environment
> (Hopper) eps=0.075|clean|random|critic|MAD|Min RS|Optimal attack
> ---|---|---|---|---|---|---
> AS-PPO (Ours)|3667|3546|**3706**|2916|1558|1500
> Radial-PPO|**3740**|**3729**|No result|3214|2141|1722
> ATLA-PPO|3487|3474|3524|3081|1567|1224
> WocaR-PPO|3616|3633|No result|**3541**|**3277**|**2390**
>
> >**Q4:** I am concerned that the literature survey of robust RL baselines by the authors might not be comprehensive, especially with regard to recent works. (The authors do cite [1], but why do not discuss ATLA?)
>
> **A4:** We use the Optimal attack proposed in ATLA [1] to evaluate our agents in the paper. We will include the ATLA baseline in Table B.1 and B.2 in the revision.
>
> >**Q5:** In Table 3, why is the comparison limited to SA-PPO and vanilla PPO, and where are Radial-PPO and other PPO-based baselines? It appears that the proposed method performs better only on larger attack budgets. However, in Table 8, it shows that AS-PPO does not have a significant advantage, or even any advantage, on large budgets.
>
> **A5:** We include an additional comparison with Radial-PPO and other works in Table B.1 and B.2. For the performance of our agents, it is true that our agents do not always outperform the previous works under every setting. However, our agents demonstrate a better trade-off between the clean reward, empirical robustness, and robust guarantee. Our agents still have a much better overall performance.

---

> ### Author Response · Authors · 2023-11-19
> **Author Respone (2/2)**
>
> >**Q6:** In Table 5, the paper fails to clearly demonstrate the advantages of DS-DQN and AS-PPO; the terms "high" or "highest" are quite vague and do not provide an intuitive representation of contributions.
>
> **A6:** Table 5 shows that our agents are the first to achieve strong empirical robustness and provable robustness guarantees at the same time. The purpose of Table 5 is to point out that the previous state-of-the-art focused on comparing the empirical robustness, while the part of certifiable agents is still without much exploration.
>
> >**Q7:** This paper introduces attack methods based on RS. However, the authors do not compare their attack method with the strongest evasion attacks, such as PA-AD[3], nor do they provide a discussion of prior attack methods in the related work.
>
> **A7:** Our attack suggests also smoothing the adversary or introducing RS in the PGD attack to enhance the attack against the smoothed agents. Our proposed attack method only makes a difference to the original attack when the victim is a smoothed agent. In fact, the RS setting can be applied to every attack, including PA-AD, to transform the given attack into an attack target at the smoothed agents. Here we give an additional analysis of how our proposed method affects the PA-AD attack in Table C. The method S-PGD and S-PA-AD denotes the smoothed attack which is our proposed method. The results show that in order to effectively reduce the reward of a smoothed agent, one should consider the smoothing variance.
>
> **Table C.** The comparison between the original attack and the attack after smoothing.
> DSDQN eps=0.05|No attack|PGD attack|PA-AD|S-PGD attack (Ours)|S-PA-AD (Ours)
> ---|---|---|---|---|---
> Pong|21.0 +- 0.00|19.4 +- 2.33|19.0 +- 2.10|**18.8 +- 1.17**|**-11.8 +- 2.79**
> Freeway|33.2 +- 0.40|30.4 +- 1.85|28.4 +- 1.36|**23.8 +- 1.17**|**1.4 +- 0.80**
> RoadRunner|35420 +- 5116|16200 +- 1482|41760 +- 5827|**0 +- 0**|**0 +- 0**
>
> **Summary**
>
> In summary, we have:
> * In Q1, explain that there are diagrams for algorithms in Appendix A2 and A3.
> * In Q2, discuss the possible limitations.
> * In Q3, Q4, and Q5, include more recent baselines.
> * In Q5, explain the advantages of our methods.
> * In Q6, explain that the goal of Table 5 is to give readers a quick idea of our advantages.
> * In Q7, include the PA-AD attack and explain that our attack setting can be also applied to PA-AD. Point out that it is important to smooth the attack while attacking the smoothed agents.
>
> We believe that we have addressed all your concerns. Please let us know if you still have any reservations and we would be happy to address them!

---

> ### Author Response · Authors · 2023-11-21
> **Request Rebuttal feedback from Reviewer RhGb**
>
> Dear Reviewer RhGb,
>
> We believe that we have addressed all your concerns with additional new experiments (see Table A.1, A.2, B.1, and B.2 in Q3, and also Table C in Q7). Please let us know if you still have any reservations and we would be happy to address them. Thank you!

---

> > ### Comment · Reviewer_RhGb · 2023-11-23
> > **Thanks to the response**
> >
> > I appreciate the response from the authors, which has led to improvements in the experiment and presentation of this paper. While I still have some concerns regarding the motivation and novelty, I have decided to raise my score.

---

> > > ### Author Response · Authors · 2023-11-23
> > > **Further clarification on the motivations and novelty**
> > >
> > > Thanks for raising the score! We would like to further address your concerns. Our motivation comes from the very low clean reward of current smoothed agents. We think that the ultimate goal of smoothed agents is to achieve a high guarantee and high reward under attack at the same time. The guarantee of smoothed agents would be meaningless given that the clean performance is severely degraded.
> > >
> > > As for the novelty, we are the first to look into the practical issue of smoothed agents and also the first to focus on achieving SOTA with strong provable guarantees, while previous work most focus on the empirical reward under attack.
> > >
> > > We also give a detailed list of our 4 major contributions in this work, hope this makes everything clear.
> > >
> > > 1. **Contributions related to the simple Randomized Smoothing (RS):**
> > >     * We identify the failure mode of the existing smoothed DRL agents, showing that the naively smoothed agents have poor trade-offs on the clean reward and robustness.
> > >     * We point out that the robustness of the CROP agents might be overestimated due to the smoothing strategy and attack they used. We fix this issue by introducing hard RS and a stronger attack.
> > >     * We extend CROP’s proposed robust guarantee for DQN agents to the PPO setting and defined action bound. To our best knowledge, the action bound for PPO has never been derived before. We also do experiments on this bound (see Appendix A.13).
> > > 2. **Contributions related to robust DRL agents:**
> > >     * Our agents achieved state-of-the-art results under attack. Following the reviewers’ suggestions, We include new baselines and show that our agent still outperforms the current best WocaR-RL in most of the environments.
> > >     * Our agent is the first state-of-the-art agent with high robustness guarantee at the same time (certified radius and reward lower bound), while the previous state-of-the-art only evaluated their agents under empirical attack.
> > > 3. **Contributions related to Randomized Smoothing (RS) in DRL:**
> > >     * We develop the first robust DRL training algorithms leveraging randomized smoothing for both discrete actions (DS-DQN) and continuous actions (AS-PPO).
> > >     * We show that simply training with RS does not work, and is necessary to use denoised smoothing and adversarial training.
> > >     * We point out that different smoothing strategies can affect the robustness guarantee. (i.e. Our hard RS strategy is not affected by the output range of the Q-network and generally achieves a larger certified radius.)
> > > 4. **Contributions related to adversarial attack:**
> > >     * We develop a new attack aiming at attacking smoothed agents. This attack model we proposed can be easily used on any attack based on optimization. We show how our setting can be applied to the PA-AD attack, denoted as S-PA-AD, and lead to a stronger attack for smoothed agents.
> > >
> > > Thanks for your valuable feedback! Please let us know if you have further concerns. We are happy to address them!

---

### Official Review · Reviewer_Jf7U · 2023-11-07

**Soundness:** 2 fair
**Presentation:** 3 good
**Contribution:** 2 fair
**Rating:** 5
**Confidence:** 3

**Summary:**

This paper proposes new algorithms to train smoothed robust DRL agent and achieve good reward. Experimental results show that the algorithms outperform existing baselines. A new adversarial attach is also proposed and is shown to be more effective in decreasing agent rewards.

**Strengths:**

- The problem in consideration is interesting and timely.

**Weaknesses:**

- It would be useful to explain how representative the issues in CROP are. Also, are there recent works that address/avoid these issues already? Fig.1 only compares results with CROP. It would be useful to have results from other methods.
- One issue I have is that the paper seems to contain two pieces of results, one for discrete and the other for continuous, and they seem to be quite orthogonal. It would be helpful to explain the common components of the schemes.
- The baselines are all before 2022. Are there more recent methods? If so, please compare with them.

**Questions:**

- It would be useful to explain how representative the issues in CROP are. Also, are there recent works that address/avoid these issues already? Fig.1 only compares results with CROP. It would be useful to have results from other methods.
- One issue I have is that the paper seems to contain two pieces of results, one for discrete and the other for continuous, and they seem to be quite orthogonal. It would be helpful to explain the common components of the schemes.
- The baselines are all before 2022. Are there more recent methods? If so, please compare with them.

---

> ### Author Response · Authors · 2023-11-19
> **Author Response**
>
> Thank you for reviewing and pointing out some main concerns.
>
> >**Q1:** It would be useful to explain how representative the issues in CROP are. Also, are there recent works that address/avoid these issues already? Fig.1 only compares results with CROP. It would be useful to have results from other methods.
>
> **A1:** To our best knowledge, there is another robust RL literature related to randomized smoothing
>
> [1] Kumar, et al. “Policy Smoothing for Provably Robust Reinforcement Learning”, ICLR 2022
>
> CROP proved bounds for randomized smoothing in RL, while [1] focused on showing that the bounds in the Supervised Learning setting cannot directly transfer to RL setting because of the non-static nature of RL. [1] provided another proof for the bounds and pointed out that the proof in CROP might have some issues. These two papers all do simple randomized smoothing during the inference. Hence, the current works of randomized smoothing for RL all fall into the issue of low clean reward when the smoothing variance increases. We consider this issue important since the agents with strong guarantees but poorly performed are unacceptable.
>
> >**Q2:** One issue I have is that the paper seems to contain two pieces of results, one for discrete and the other for continuous, and they seem to be quite orthogonal. It would be helpful to explain the common components of the schemes.
>
> **A2:** In this work, we focused on how to effectively train a smoothed agent that is provably robust. The DQN and PPO parts both leverage randomized smoothing during the training to solve the issue of low clean reward. We feel that it is necessary to show that no matter in the discrete or continuous setting, our framework achieves high clean reward, high robust reward, and strong guarantees simultaneously.
>
> >**Q3:** The baselines are all before 2022. Are there more recent methods? If so, please compare with them.
>
> **A3:** Following your suggestion, we have conducted additional experiments in the rebuttal to provide the comparison of two additional baselines, ATLA [2] and WocaR [3] in Tables A.1, A.2, B.1, and B.2. ATLA [2] trained their agents using adversarial training with the optimal attack they proposed, which is only applicable in the PPO setting. WocaR [3] performed worst-case-aware policy optimization without using an adversary and can be applied to both DQN and PPO settings.
>
> The DQN results are in Table A.1 and A.2. We found that WocaR-DQN can only tolerate a smaller attack budget as the previous works did, which is not comparable to our DS-DQN (Vanilla) without using any robust base agent.
>
> The PPO results are in Table B.1 and B.2. ATLA is generally not comparable to our AS-PPO in both Walker and Hopper environments. WocarR outperforms our AS-PPO in the Hopper environment, while is less robust than AS-PPO under the Walker environment.
>
> We would like to highlight that all these baselines do not come with robustness guarantees like our agents. Our agents are not only empirically robust but also provably robust.
>
> **Table A.1.** The reward of DS-DQN and WocaR-DQN under PGD attack in the Pong environment
> (Pong) eps|0.01|0.02|0.03|0.04|0.05
> ---|---|---|---|---|---
> Our DSDQN (Vanilla)|19.2+-0.75|**18.4+-2.15**|**19.2+-1.17**|**17.2+-2.56**|**18.8+-1.17**
> WocaRDQN|**21.0+-0.0**|-21.0+-0.0|-21.0+-0.0|-21.0+-0.0|-21.0+-0.0
>
> **Table A.2.** The reward of DS-DQN and WocaR-DQN under PGD attack in the Freeway environment
> (Freeway) eps|0.01|0.02|0.03|0.04|0.05
> ---|---|---|---|---|---
> Our DSDQN (Vanilla)|**32.6+-1.20**|**31.6+-1.50**|**30.0+-1.10**|**28.0+-1.41**|**23.8+-1.17**
> WocaRDQN|21.4 +- 1.36|21.8 +- 1.17|21.0 +- 0.89|21.0 +- 0.63|21.0 +- 1.10
>
> **Table B.1.** The reward of AS-PPO, ATLA-PPO, and WocaR-PPO under attacks in the Walker environment
> (Walker) eps=0.05|clean|random|critic|MAD|Robust Sarsa|Optimal attack
> ---|---|---|---|---|---|---
> AS-PPO (ours)|**4969**|**5039**|**5488**|**5290**|**4323**|**4296**
> ATLA-PPO|3920|3779|3915|3963|3219|3463
> WocaR-PPO|4156|4244|No result|4177|4093|3770
>
> **Table B.2.** The reward of AS-PPO, ATLA-PPO, and WocaR-PPO under attacks in the Hopper environment
> (Hopper) eps=0.075|clean|random|critic|MAD|Robust Sarsa|Optimal attack
> ---|---|---|---|---|---|---
> AS-PPO (ours)|**3667**|3546|**3706**|2916|1558|1500
> ATLA-PPO|3487|3474|3524|3081|1567|1224
> WocaR-PPO|3616|**3633**|No result|**3541**|**3277**|**2390**
>
> [2] Zhang, et al. “Robust Reinforcement Learning on State Observations with Learned Optimal Adversary”, ICLR 2021
>
> [3] Liang, et al. “Efficient Adversarial Training without Attacking: Worst-Case-Aware Robust Reinforcement Learning”, NeurIPS 2022
>
> **Summary**
>
> In summary, we have:
> * In Q1, explain that the issue in CROP also happens in other papers using RS for RL.
> * In Q2, explain the connection between the DS-DQN and AS-PPO.
> * In Q3, include more recent baselines and compare the performance.
>
> Please let us know if you still have any concerns and we would be happy to address them!

---

> ### Author Response · Authors · 2023-11-21
> **Request Rebuttal feedback from Reviewer Jf7U**
>
> Dear Reviewer  Jf7U,
> We believe that we have addressed all your concerns with additional new experiments (see Table A.1, A.2, B.1, and B.2 in Q3). Please let us know if you still have any reservations and we would be happy to address them. Thank you!

---

> > ### Comment · Reviewer_Jf7U · 2023-11-22
> > **Thank you**
> >
> > I thank the authors for their responses. However, it is still not clear to me how significant the problem is and how novel the methods are, especially since the paper appears to be mainly focusing on addressing issues of one particular prior work. Thus, I will keep my score. Thanks.

---

> ### Author Response · Authors · 2023-11-23
> **Further clarification on our contributions and novelty of this paper**
>
> Thanks for the reply and sharing your concerns! Sorry for making you feel unclear about how significant the existing problem is. We feel that the very low clean reward of the current smoothed agents is a severe issue since the goal of proposing smoothed agents is to achieve a high guarantee and high reward under attack. The guarantee of smoothed agents would be meaningless given that the clean performance is severely degraded.
>
> Our agents not only achieve high clean reward and robust reward but also provide a strong guarantee that previous SOTAs fail to achieve. Hence, we believe that our contribution is not minor and can open a new direction of further strengthening the robustness guarantee for future works.
>
> If you still feel unclear about our contribution, here we give a detailed list of our 4 major contributions in this work:
>
> 1. **Contributions related to the simple Randomized Smoothing (RS):**
>     * We identify the failure mode of the existing smoothed DRL agents, showing that the naively smoothed agents have poor trade-offs on the clean reward and robustness.
>     * We point out that the robustness of the CROP agents might be overestimated due to the smoothing strategy and attack they used. We fix this issue by introducing hard RS and a stronger attack.
>     * We extend CROP’s proposed robust guarantee for DQN agents to the PPO setting and defined action bound. To our best knowledge, the action bound for PPO has never been derived before. We also do experiments on this bound (see Appendix A.13).
> 2. **Contributions related to robust DRL agents:**
>     * Our agents achieved state-of-the-art results under attack. Following the reviewers’ suggestions, We include new baselines and show that our agent still outperforms the current best WocaR-RL in most of the environments.
>     * Our agent is the first state-of-the-art agent with high robustness guarantee at the same time (certified radius and reward lower bound), while the previous state-of-the-art only evaluated their agents under empirical attack.
> 3. **Contributions related to Randomized Smoothing (RS) in DRL:**
>     * We develop the first robust DRL training algorithms leveraging randomized smoothing for both discrete actions (DS-DQN) and continuous actions (AS-PPO).
>     * We show that simply training with RS does not work, and is necessary to use denoised smoothing and adversarial training.
>     * We point out that different smoothing strategies can affect the robustness guarantee. (i.e. Our hard RS strategy is not affected by the output range of the Q-network and generally achieves a larger certified radius.)
> 4. **Contributions related to adversarial attack:**
>     * We develop a new attack aiming at attacking smoothed agents. This attack model we proposed can be easily used on any attack based on optimization. We show how our setting can be applied to the PA-AD attack, denoted as S-PA-AD, and lead to a stronger attack for smoothed agents.
>
> Thanks for your valuable feedback! Please let us know if you have further concerns. We are happy to address them!

---

### Meta-Review · Area_Chair_JXWy · 2023-12-12

**Metareview:**

This paper studies how to improve the robustness of deep reinforcement learning (DRL) agents while preserving their utility.  It is good that the authors introduce two innovative algorithms, DS-DQN and AS-PPO, tailored to train DRL agents that achieve high clean rewards and robustness certification in both discrete and continuous action spaces. Also, DS-DQN and AS-PPO surpass previous robust DRL agents and they introduce a more potent adversarial attack, which is also the strength of this paper.

However, all the reviewers agree that it is not sufficient for this paper to only compare with CROP, as one single baseline hurts the significance and novelty of the contribution claimed in this paper. The authors should make more experiments and clarifications to show the significance of their proposed solution, which should not be a simple correction of CROP but a solution for a common issue in the previous literature.

In the current form, I tend to recommend rejection.

**Justification For Why Not Higher Score:**

See the above comments about the justification of significance.

**Justification For Why Not Lower Score:**

N/A

---

### Decision · Program_Chairs · 2024-01-16

Reject